# Semi-Supervised Single Domain Generalization with Label-Free Adversarial Data Augmentation

**Ronghang Zhu**                                                    *ronghangzhu@uga.edu*
*University of Georgia*

**Xiang Yu**                                                        *yuxiang03@gmail.com*
*Amazon*

**Sheng Li**                                                        *shengli@virginia.edu*
*University of Virginia*

**Reviewed on OpenReview:** *https://openreview.net/forum?id=sUlbRfLijj*

## Abstract

Domain generalization (DG) has attracted increasing attention recently, as it seeks to improve the generalization ability of visual recognition models to unseen target domains. DG leverages multiple source domains for model training, while single domain generalization (SDG) further restricts such setting by exploiting only a single source domain. Nevertheless, both DG and SDG assume that the source domains are fully labeled, which might not be practical in many real world scenarios. In this paper, we present a new problem, i.e., semi-supervised single domain generalization (SS-SDG), which aims to train a model with a partially labeled single source domain to generalize to multiple unseen testing domains. We propose an effective framework to address this problem. In particular, we design a label-free adversarial data augmentation strategy to diversify the source domain, and propose a novel multi-pair FixMatch loss to generalize classifiers to unseen testing domains. Extensive experiments on OfficeHome, PACS and DomainNet20 datasets show that our method surpasses the latest SDG and semi-supervised methods. Moreover, on PACS and DomainNet20, our method approaches the fully supervised ERM upper bound within 5% gap, but only uses less than 8% of the labels.

## 1 Introduction

Deep neural networks (DNNs) lead large success in the past decade in many fields, e.g., object detection and classifications. Many of the applications rely on the assumption that training and testing distributions are identical or close. However, in real scenarios, data acquiring always encounters the environment variance, i.e., the lighting changes from dawn to night, or the camera moves from one place to another. The environment variance inevitably brings in the domain shift for the captured training and testing data (Recht et al., 2019; Hendrycks & Dietterich, 2019). Closing this domain discrepancy has become one of the recent popular topics in the community.

Domain adaptation (DA) (Wang & Deng, 2018) and domain generalization (DG) (Zhou et al., 2021a; Wang et al., 2021a) are the major techniques to tackle this problem. DA methods jointly exploit the source and target domain data for model training, in which the methods attempt to align the feature space between the source and target domains. While DG methods solve a more challenging task, utilizing multiple labeled source domain data to learn towards a generalized model, to predict the target domain data which is unavailable in the training process. Compared to DA, DG relaxes the assumption on target domains and usually enjoys better model generalization ability. The differences between DA and DG are illustrated in Figure 1.

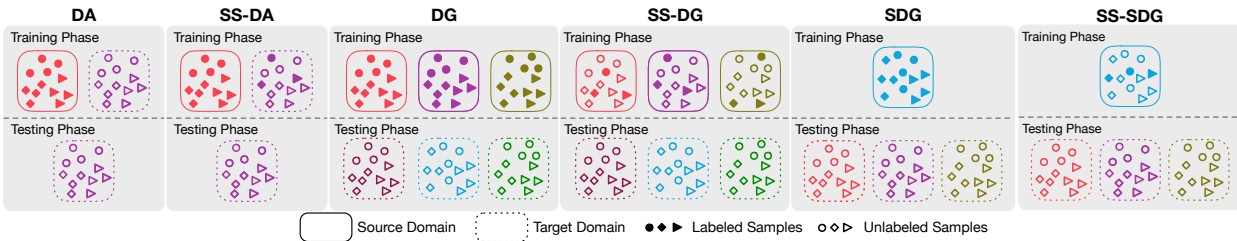

Figure 1: Problem setting differences among domain adaptation (DA), single domain generalization (SDG), domain generalization (DG), semi-supervised domain adaptation (SS-DA), semi-supervised domain generalization (SS-DG), single domain generalization (SDG), and our newly introduced setting semi-supervised single domain generalization (SS-SDG) task. Different colors indicate different domains.

Though promising, DG still faces two major limitations. First, DG methods require multiple source domains for model training. In practice, it would be expensive or even infeasible to collect multiple source domains' data. To address this problem, single domain generalization (SDG) (Qiao et al., 2020) has been proposed recently. Different from DG, SDG aims to train a model with a single source domain data and generalize to multiple unseen target domains. Second, DG methods require that the source domains should be fully labeled, which is usually expensive and labor intense. However, in real-world application scenarios, it's not always feasible to have access to a fully labeled source domain. Using face recognition as an example, the offline face recognition model relies on labeled data typically obtained in controlled environments or through prior registration processes. However, in dynamic and open settings like surveillance scenarios, encountering new users may not involve a prior registration step, resulting in a significant portion of unlabeled data. In this context, the term "source" data remains consistent, representing the labeled dataset acquired during the offline phase. Conversely, the "target" data show variability across diverse scenarios, including indoor and outdoor environments. This discrepancy poses a significant challenge since the offline data are well-labeled, while real-time data, forming the "target" data, are mostly unlabeled. In essence, the practical application of SS-SDG in face recognition grapples with effectively managing both labeled offline data and predominantly unlabeled real-time data. The overarching goal is to develop a model that is both generalizable and robust for face recognition. Very recently, a solution to this challenge has emerged in the form of semi-supervised domain generalization (SS-DG) (Zhou et al., 2021b). This approach operates under the assumption that only a small portion of the samples are labeled in the source domains. Clearly, SDG and SS-SDG separately tackle each of the two limitations, i.e., single source domain and few labeled data. Different from existing problems, we desire a unified framework that can boost domain generalization from both of the two aspects.

Consequently, we propose a more practical yet unsolved problem, i.e., semi-supervised single domain generalization (SS-SDG). In this problem, we assume that only one source domain is available for model training, and it consists of a few labeled and abundant unlabeled samples. The relationships between SS-SDG and other related problems are illustrated in Figure 1. While SDG is the most relevant setting to ours, most of existing SDG methods (Zhao et al., 2020; Qiao et al., 2020; Fan et al., 2021) are based on adversarial data augmentation (Volpi et al., 2018) and require the label information to generate new samples to enrich the diversity of source domain. As the result, these SDG methods would fail in our proposed SS-SDG setting which cannot provide sufficient and accurate label information for data augmentation. Our empirical results validate that existing SDG methods perform unsatisfying in our SS-SDG task.

To address the new challenging problem, we propose a novel label-free adversarial data augmentation framework to enrich the source domain diversity without label information, as well to leverage such generated data in a multi-pair FixMatch way to regularize for better training. Inspired by self-supervised learning (SSL) (Yang et al., 2021), we design a label-free adversarial data augmentation strategy, which is an interactive feature extractor pre-training and adversarial sample generation approach without label information, to enhance the diversity of source domain data. Given source domain and the newly generated samples, we organize them into multiple training pairs and propose a novel multi-pair FixMatch (MPFM) loss to regularize the classifier training for better generalization ability to unseen testing domains.

Our contributions are thus summarized as:

- We introduce a new challenging domain generalization task, namely the semi-supervised single domain generalization (SS-SDG), addressing the scenario of training on a partially labeled single source domain while generalizing to multiple unseen target domains.

- We propose an efficient framework, introducing a novel label-free adversarial data augmentation strategy to enrich diversity of the single source domain diversity without label information, and a multi-pair FixMatch regularization to better utilize the diversified data towards a more generalized classifier for unseen target domains.

- We conduct extensive SS-SDG experiments on OfficeHome, PACS and DomainNet20, and achieve superior performance over the state-of-the-arts, e.g., approaching supervised ERM upper bound within 5% accuracy gap by only using less than 8% labels.

## 2 Related Work

### 2.1 Domain Adaptation

The basic idea of domain adaptation (DA) (Zhu et al., 2023; Shi et al., 2022) is to leverage the labeled source data and unlabeled target data to adapt the model from the source domain to target domain. Existing domain adaptation methods address the domain gap between source and target domains via distribution alignment by statistical metrics (Gretton et al., 2006; Shen et al., 2018) or adversarial learning (Ganin & Lempitsky, 2015; Zhu et al., 2021b;a; Zhu & Li, 2022b; Rezayi et al., 2023). Among different DA problem settings (Zhu & Li, 2021), Semi-supervised domain adaptation (SS-DA) (Donahue et al., 2013; Saito et al., 2019) is the most relevant DA problem to our proposed SS-SDG where a few target labels are available. The early SS-DA efforts (Donahue et al., 2013; Yao et al., 2015; Ao et al., 2017) involved the creation of individual models for each domain by regulating through the use of constraints. Recently, the Minimax Entropy (MME) (Saito et al., 2019) utilizes the conditional entropy value of unlabeled target data to play min-max game between feature encoder and classifier to minimize domain discrepancy and simultaneously acquire discriminative features for the task. AdaMatch (Berthelot et al., 2021) induces random logit interpolation to address the differences between source and target domains. Domain adaptation is indeed a potential solution for our proposed SS-SDG problems. However, the requirement of having access to the target domain data hinder the application of traditional domain adaptation methods to our SS-SDG challenges.

### 2.2 Domain Generalization

Different from domain adaptation, domain generalization (DG) (Wang et al., 2021a; Zhou et al., 2021a) takes into account situations where the target domain is not available during the process of model learning. The goal of DG is to learn a generalized model with multiple source domains to adapt to an unseen target domain. The DG methods can be roughly divided into four groups, i.e., domain alignment based methods (Ganin et al., 2016; Li et al., 2018b; Piratla et al., 2020) , meta-learning based methods (Li et al., 2018a; Dou et al., 2019; Du et al., 2020), data augmentation based methods (Shankar et al., 2018; Zhou et al., 2020a; Robey et al., 2021; Huang et al., 2021), and casual based methods (Dai et al., 2023; Mahajan et al., 2021). For domain alignment based methods, the central concept is to minimize the discrepancy between source domains for learning domain-invariant representations. The Maximum Mean Discrepancy based Adversarial Autoencoder (MMD-AAE) (Li et al., 2018b) minimizes the MMD distance between source domains in feature space, while employing adversarial learning to make the distributions of the source domains similar to a prior distribution. For meta-learning based methods, the main idea is to leverage the multiple source domains in a meta-train and meta-test manner and update the gradient in a more sophisticated way. The Meta-Learning Domain Generalization (MLDG) (Li et al., 2018a) proposes a model-agnostic training procedure to improve the generalization of learned model by simulating the train/test domain shift during training by generating virtual test domains. For data augmentation based methods, the goal is to explore how to generate extra synthetic data based on the source domain data, and use the joint data to recover the unseen target domain distribution. The Model-Based Domain Generalization (Robey et al., 2021) proposes a novel framework

for DG, in which the goal is to impose invariance to the underlying transformations of data that reflect the variations between domains. For causal-based methods, the objective is to employ causal reasoning to establish invariant conditions for domain generalization. An example of this approach is MatchDG (Dai et al., 2023), which introduces an object-invariant condition for domain generalization. This is achieved by minimizing the distance between representations of the same object across different domains through causal reasoning. However, these existing domain generalization (DG) methods are not well-suited for addressing the SS-SDG problem we have proposed. The primary reason is that they depend on having access to fully labeled multi-source domains, making them unsuitable for direct application to the SS-SDG scenario.

## 2.3 Single Domain Generalization

Single Domain Generalization (SDG) is a more challenging setting compared to domain generalization, which only uses one source domain to learn a model and generalize to multiple unseen target domains. The early SDG method, i.e., Meta-Learning based Adversarial Domain Augmentation (M-ADA) (Qiao et al., 2020), jointly utilizes adversarial data augmentation (Volpi et al., 2018) relaxed by a Wasserstein auto-encoder and meta-learning to improve the diversity of generated samples and the generalization of learned model. Following this direction, numerous methods (Zhao et al., 2020; Li et al., 2021; Fan et al., 2021; Wang et al., 2021b; Zhu & Li, 2022a) have been proposed for expanding the source data by generating samples that fall outside the source domain, with the goal of encompassing the range of the target domain. For examples, the Learning-to-Diversify (L2D) (Wang et al., 2021b) proposes a style-complement module to improve the model's generalization ability. This is achieved by synthesizing samples from diverse distribution that complement the original source domain. Further, two-step wise mutual information optimization is used to learn distinctive features from diversely styled data. The Progressive Domain Expansion Network (PDEN) (Li et al., 2021) designs two subnetworks, i.e., domain expansion subnetwork and representation learning subnetwork, to work together in a mutually beneficial way through joint learning. The former subnetwork aims to generate multiple domains progressively to simulate various photometric and geometric transformations in unseen domains. The later subnetwork tries to learn a domain invariant representation that clusters each class well, resulting in a better decision boundary and improved generalization. Recently, the Center-aware Adversarial Domain Augmentation (CADA) (Chen et al., 2023) proposes a novel angular center loss to modify the source samples in such a way as to move them away from the class centers. Thus, the generated samples are diversified enough to simulate a large domain shift and imporve the generalization of learned model. In contrast to standard single domain generalization (SDG), we introduce a more practical yet challenging setting known as semi-supervised single domain generalization (SS-SDG), where the availability of labeled samples in the source domain is limited. Unlike traditional SDG methods, which heavily rely on labeled data for data augmentation, our SS-SDG setting presents a unique challenge. Here, the effectiveness of these SDG methods may be compromised due to the insufficient availability of labeled information in our setting.

## 2.4 Semi-Supervised Learning

Semi-Supervised Learning (SSL) is a fundamental research topic in computer vision and machine learning, which seeks to learn a model with few labeled and a large amount of unlabeled data. SSL methods can be generally divided into three categories: (1) The pseudo-labeling based methods (Lee et al., 2013; Berthelot et al., 2020; Xie et al., 2020b), utilize the intermediate model to predict the pseudo ground truth label and iteratively update the model with the pseudo labeled data. For example, the MixMatch (Berthelot et al., 2020)induces distribution alignment and augmentation anchoring to simultaneously align the marginal distribution of prediction on unlabeled data with the marginal distribution groundtruth labels and enforce the output of multiple highly transformed versions of same data to be similar for a weakly-transformed version of the data. (2) The consistency constraint (Oliver et al., 2018) based methods (Tarvainen & Valpola, 2017; Miyato et al., 2019; Zhang & Qi, 2020), leverage the consistency across the same data with multiple augmentations or perturbed models to regularize the embedding learning. For instance, the Worse-Case Perturbation (WCP) (Zhang & Qi, 2020) is inspired by the concept that a robust model should not be easily impacted by minor variations and keep the output decision remain as consistent as possible for both label and unlabeled data. Therefore, two regularizations, i.e., additive and DropConnect perturbations, are proposed to perturb the model by injecting noise to model weights and changing connections in model structure.

(3) The comprehensive methods (Sohn et al., 2020; Xie et al., 2020a; Zhang et al., 2021) combine both the pseudo-labeling and data augmentation for more performance boost. Such as, the FlexMatch (Zhang et al., 2021) proposes a curriculum learning strategy to make use of unlabeled data based on the model's learning progress. The essence of this strategy is to dynamically adjust thresholds for different classes at each iteration, allowing informative unlabeled data and their pseudo labels to be utilized. The canonical SSL setting would be impractical, as it assumes the training distribution and testing distribution are similar. To relax the assumption, a new semi-supervised domain generalization (SS-DG) (Zhou et al., 2021b) is proposed, i.e., StyleMatch (Zhou et al., 2021b), which tackles SS-DG by inducing style augmentation (Huang & Belongie, 2017a) to extend source domain distribution in the FixMatch (Sohn et al., 2020) framework, and alleviates overfitting problem with stochastic classifier. Our proposed approach, semi-supervised single domain generalization (SS-SDG), represents a significant advancement in the field. It deals with the challenge of generalizing from a single source domain and a limited amount of labeled data to multiple target domains. In this context, traditional semi-supervised domain generalization (SS-DG) methods, such as FixMatch, tend to perform sub-optimally due to the extreme scarcity of labeled data available in the source domain.

## 2.5 Data Augmentation

Data augmentation has become a widespread practice for enhancing the training of machine learning models, helping to mitigate overfitting and improve generalization. The fundamental concept behind data augmentation involves extending the original (data, label) pairs with new pairs, denoted as ($\mathbb{T}$(data), label), where $\mathbb{T}(\cdot)$ represents a transformation. Existing data augmentation methods can be roughly categoried into three categories: (1) Hand-crafted transformations: the main idea is to encompass the stacking of traditional image transformations within $\mathbb{T}$, such as random flips, random crops, and rotations. In addition to these conventional image transformations, there have been explorations into techniques like Mixup (Zhang et al., 2017), which involves mixing instances at the image level. However, it's important to note that these transformations may have limitations in certain applications, potentially altering the associated labels. (2) Adversarial learning based augmentation: inspired by adversarial attacks (Xu et al., 2020a), this branch involves perturbing images using sign-flipped gradients back-propagated from a classifier. This method is commonly used in single domain generalization to generate "challenging" images, making it more difficult for the classifier to make accurate judgments. However, a drawback is the increased computational cost associated with these techniques. (3) CNN based augmentation: in this category, Convolutional Neural Networks (CNNs) (Gu et al., 2018) serve as the transformation function $\mathbb{T}(\cdot)$. The primary goal is to alter the styles of images, thereby enriching the diversity of the existing dataset. A notable limitation of these learnable image synthesis methods is that the newly generated images may not exhibit significant differences from images in other domains due to the lack of style information. This direction can be further divided into two subcategories. The first one is learnable based augmentation (Yue et al., 2019; Zhou et al., 2020b) which focuses on learning an image translation model or employing pre-existing style transfer models like CycleGAN (Zhu et al., 2017) or AdaIN (Huang & Belongie, 2017b). While the other one is Non-learnable augmentation (Xu et al., 2020b) where $\mathbb{T}(\cdot)$ is randomly initialized with a Gaussian distribution at each iteration. While CNN-based augmentation is a potential solution for addressing generalization tasks related to a single source domain, it's important to note that its requirement of multiple source domains makes it less applicable to the SS-SDG problem, which primarily deals with a single source domain. In contrast, adversarial learning-based augmentation has demonstrated its efficiency in tackling single domain generalization (SDG) problems. Therefore, opting for an adversarial learning-based approach represents a promising and practical solution for addressing SS-SDG challenges.

## 3 Our Approach

The overall proposed framework is illustrated in Fig. 2 which consists of two stages. At stage 1, we designed two phases: pretrain phase and data generation phase. In pretrain phase, we adopt contrastive learning to adapt the feature generator **G** to source domain by mining the similarity among samples in feature space. In data generation phase, we generate new diversified samples to complement source domain by perturbing source samples in an adversarial way with contrastive learning. The data generation phase is alternated with the pretrain phase. At stage 2, we jointly train the classifier **C** and the pre-trained **G** by adopting

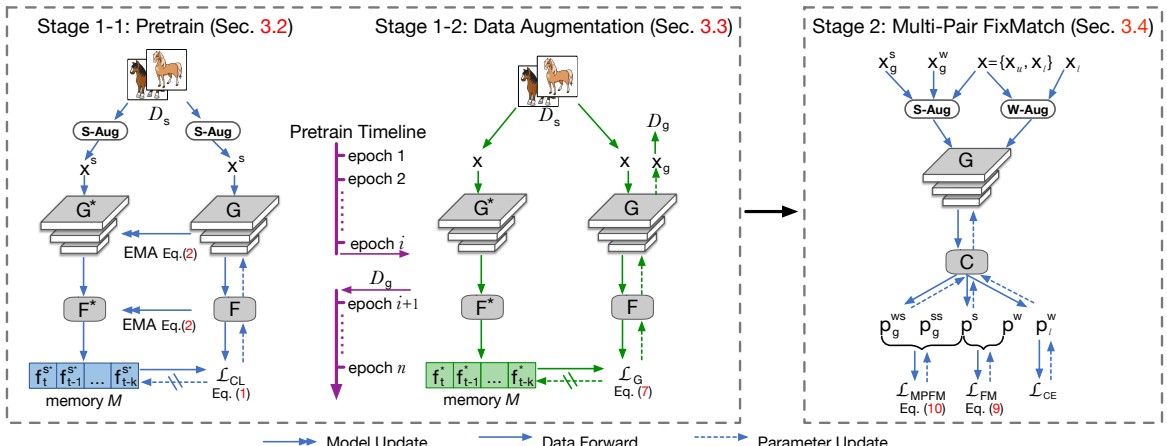

Figure 2: The proposed framework for SS-SDG. Our framework consists of two stages. At stage 1, we jointly train a feature extractor **G** in a self-supervised manner by excavating samples intrinsic supervision with contrastive learning (Stage 1-1), and generate new samples to enrich the diversity of source domain in an adversarial way (Stage 1-2) without label information. At stage 2, We propose a novel multi-pair FixMatch (MPFM) loss to better utilize the rich pair information for better generalization on unseen target domains.

cross-entropy loss with source labeled samples. We also propose a novel loss function, Multi-Pair FixMath (MPFM) in Eq. equation 10, to propagate useful training signals from source unlabeled samples and generated samples. Firstly, MPFM generates pesudo-labels utilizing the classifier **C**'s output on weakly transformed unlabeled images. These pesudo-labels will be treated as true labels for a strongly transformed version of same sample to train **C** and **G** with cross-entropy loss. Here the weak transformation indicates a simple data augmentation which includes a random cropping and horizontal flipping while strong transformation denotes extensive data augmentation like RandAugment (Cubuk et al., 2020) and CTAugment (Berthelot et al., 2019). The full algorithm is summarized in Algorithm 1.

Below, we introduce the framework in detail. Firstly, We start by introducing the SS-SDG problem with formal notations in Sec. 3.1. Then, we introduce the model pre-train (Figure 2 "Stage 1-1") in Sec. 3.2 including the contrastive learning loss design and the model parameter update. Further, we illustrate the new label-free adversarial data augmentation (Figure 2 "Stage 1-2") in Sec. 3.3. Finally, we explain our novel multi-pair FixMatch loss (Figure 2 "Stage 2") in Sec. 3.4 and propose the overall training objective.

## 3.1 Preliminaries

In semi-supervised single domain generalization (SS-SDG), a source domain is denoted as $\mathcal{D}_s = \{\mathcal{D}_s^l, \mathcal{D}_s^u\}$, where $\mathcal{D}_s^l = \{(x_s^{l,i}, y_s^i)\}_{i=1}^{m_s^l}$ is the portion of limited labeled samples and $\mathcal{D}_s^u = \{(x_s^{u,i})\}_{i=1}^{m_s^u}$ indicates abundant unlabeled samples. The multiple target domains are defined as $\mathcal{D}_t = \{\mathcal{D}_t^1, ..., \mathcal{D}_t^n\}$ where $\mathcal{D}_t^i = \{(x_{t,k}^i)\}_{k=1}^{m_t^i}$ indicates the data from the $i$-th unseen target domain. The source and target domains are sampled from different distributions but share the same label space. Our goal is to learn a model $\{\mathbf{G}, \mathbf{C}\}$, that can perform well on unseen multiple target domains $\mathcal{D}_t$ by utilizing partially labeled single source domain $\mathcal{D}_s$. In particular, we aim to learn a feature extractor $\mathbf{G} : x \to g$ that maps the input sample $x$ into an embedding space, and train a classifier, $\mathbf{C} : g \to p$ that conducts classification by optimizing:

$$\min \frac{1}{n} \sum_{i=1}^n \mathbb{E}_{x \in \mathcal{D}_t^i} \mathcal{L}(\mathbf{C}(\mathbf{G}(x)), y) \tag{1}$$

where $\mathcal{L}$ is a general classification loss term. Notice that $y$ is known only at the testing phase for the target domains. **G** and **C** are learned only with the source domain data. We hereby introduce the learning process.

## 3.2 Self-supervision with Contrastive Learning

Previous SDG methods (Volpi et al., 2018; Zhao et al., 2020) may encounter degraded performance in the SS-SDG setting as the source domain data is with insufficient labels. Targeting on the lacking label venue, recent contrastive learning (He et al., 2020; Chen et al., 2020; Grill et al., 2020; Zbontar et al., 2021; Zhu et al., 2021c) methods have shown promising results on unsupervised feature learning. They train the feature extractor by pulling the features from same sample under different augmented views close, and pushing the features from different samples apart.

Following the spirit, we propose to adopt contrastive learning framework, i.e., MoCo (He et al., 2020) to pre-train the feature extractor $\mathbf{G}$ with all source samples. As shown in Figure 2 "Stage 1-1: Pretrain", the contrastive framework consists of two feature extractors $\mathbf{G}^*$ and $\mathbf{G}$, two MLP projection heads $\mathbf{F}^*$ and $\mathbf{F}$, and a memory bank $M$ used to store recent $k$ samples' features by adopting the First-In-First-Out strategy to dynamically update it. Notice that $\mathbf{F}$ and $\mathbf{F}^*$ are the MLP projection heads only for the pre-train stage. They are different from our final classifier $\mathbf{C}$. We apply strong transformation (i.e., RandAugment (Cubuk et al., 2020)) on source sample $x$ twice, to obtain two strongly transformed inputs. Then by feeding these two inputs into $\mathbf{G}$ and $\mathbf{G}^*$, respectively, we compute the similarity between the transformed sample feature $f_t^s$ and all features stored in memory bank and adopt the InfoNCE (Oord et al., 2018) loss to optimize the framework:

$$\mathcal{L}_{CL} = -\log \frac{\exp\left(f_t^s \cdot f_t^{s^*}/\tau\right)}{\sum_{i=0}^{k} \exp\left(f_t^s \cdot f_{t-i}^{s^*}/\tau\right)}, \tag{2}$$

where $f_t^s$ represents the features of the $t$-th iteration inputs, and $\tau$ is a temperature hyper-parameter that controls the concentration level of the distribution (Hinton et al., 2015). To improve the consistency of features in memory bank, we employ the exponential moving average (EMA) strategy (Cai et al., 2021), which could improve the generalization of the learned model, to update parameters ($\theta^*$) and normalization factors ($\mu^*$ and $\sigma^*$) in $\mathbf{G}^*$ and $\mathbf{F}^*$ from $\mathbf{G}$ and $\mathbf{F}$:

$$\begin{aligned} \theta^* &\leftarrow \alpha\theta^* + (1-\alpha)\theta, \\ \mu^* &\leftarrow \alpha\mu^* + (1-\alpha)\mu, \\ \sigma^* &\leftarrow \alpha\sigma^* + (1-\alpha)\sigma, \end{aligned} \tag{3}$$

where $\alpha$ is a momentum coefficient close to 1, e.g., 0.999. The parameters $\theta$ in $\mathbf{G}$ and $\mathbf{F}$ are learned by standard SGD optimizer. There is no back-propagation through the $\mathbf{G}^*$ and $\mathbf{F}^*$. The $\mathbf{G}^*$ and $\mathbf{F}^*$ can be viewed as the smooth temporal ensemble of $\mathbf{G}$ and $\mathbf{F}$ along the training iterations.

## 3.3 Label-Free Adversarial Data Augmentation

While the self-supervised pre-train provides a good initialization, the problem of generalization to unseen target domains still exits. As we face the challenge of only using a single source domain to train $\mathbf{G}$ and $\mathbf{C}$, the thumb obstacle is the source data diversity. A highly concentrated source distribution can easily cause model overfitting. Many existing single domain generalization (SDG) methods (Volpi et al., 2018; Zhao et al., 2020; Qiao et al., 2020; Fan et al., 2021; Wang et al., 2021b) adopt the adversarial data augmentation fashion (Goodfellow et al., 2015) to complement the diversity of source domain. They formulate the SDG problem into a worst-case scenario (Sinha et al., 2018):

$$\min_{\psi} \sup_{\mathcal{D}_t} \left\{ \mathbb{E}[\mathcal{L}_{ce}(\psi; \mathcal{D}_t) : d(\mathcal{D}_t, \mathcal{D}_s) \leq \rho] \right\}, \tag{4}$$

where $d$ represents a distance metric to evaluate the distribution similarity between source and target domains. $\rho$ indicates the largest domain discrepancy between $\mathcal{D}_s$ and $\mathcal{D}_t$ in embedding space. $\psi$ denotes model parameters optimized by cross-entropy loss $\mathcal{L}_{ce}$ with label information. The worst-case scenario (Equation 4) can be reformulated into a Lagrangian optimization problem with a fixed penalty parameter $\beta$:

$$\min_{\psi} \sup_{\mathcal{D}_g} \left\{ \mathbb{E}[\mathcal{L}_{ce}(\psi; \mathcal{D}_g)] - \beta d_W(\mathcal{D}_g, \mathcal{D}_s) \right\}, \tag{5}$$

where $\mathcal{D}_g$ indicates the generated domain from $\mathcal{D}_s$ and $d_W$ denotes the Wasserstein metric (Volpi et al., 2018) applied to preserve the semantics of the generated samples. The overall loss function is formulated as:

$$\mathcal{L}_{SDG}(\psi; \mathcal{D}_s) = \mathcal{L}_{ce}(\psi; \mathcal{D}_g) - \beta d_W(\mathcal{D}_g, \mathcal{D}_s), \tag{6}$$

The new domain $\mathcal{D}_g$ is generated from $\mathcal{D}_s$ by maximizing $\mathcal{L}_{SDG}$ with label information under a small number of iterations: $x_{j+1} \leftarrow x_j + \eta \nabla_{x_j} \mathcal{L}_{SDG}(\psi; x_j)$.

Clearly, existing adversarial data augmentation based methods in SDG highly rely on label information used by $\mathcal{L}_{ce}$, whereas the limited labeled source data from SS-SDG setting impede the above SDG methods to generate plentiful new samples to diversify the source data distribution. By seamlessly combining the self-supervised signal introduced in Equation 2 and the worst-case scenario in Equation 5, we formulate our label-free adversarial data augmentation as:

$$\mathcal{L}_G(\theta; \mathcal{D}_s) = \mathcal{L}_{CL}(\theta; \mathcal{D}_s) - \beta d_W(\mathcal{D}_g, \mathcal{D}_s). \tag{7}$$

Following the same updating rule, as shown in Figure 2 "Stage 1-2", we input each source sample into $\mathbf{G}^*$ and $\mathbf{G}$, respectively. We compute the similarity between the one passes $\mathbf{G}$ and all features stored in memory bank (a different memory bank from the pre-train stage). With a small number of iterations to maximize Equation 7 by:

$$x_{j+1} \leftarrow x_j + \eta \nabla_{x_j} \mathcal{L}_G(\theta; x_j) \tag{8}$$

In this way, we generate a new domain $\mathcal{D}_g$ to diversify source domain $\mathcal{D}_s$ without label information. The data augmentation phase is alternated with the pre-train phase along the training epochs.

## 3.4 Multi-Pair FixMatch Regularization

With the augmented $\mathcal{D}_g$ and original $\mathcal{D}_s$, there are rich information amongst the source domain data now. For instance, we can pair the strongly transformed data with the original data, or strongly transformed data with the weakly transformed data. Furthermore, we have the option to pair the generated data, both strongly and weakly transformed generated data, with the original data. This multi-pair approach significantly enhances our capacity to regulate feature representation learning when compared to the traditional practice of pairing only original data with transformed data. Therefore, it could introduce more diversity into the original domain. As FixMatch (Sohn et al., 2020) applies similar strong/weak transformations as in our framework, and demonstrates state-of-the-art self-supervised training performance, we consider to formulate our multi-pair constraint into a consistent format, termed Multi-Pair FixMatch regularization.

Specifically, for strongly transformed data $\mathcal{D}_s^s = Trans_s(\mathcal{D}_s)$ and weakly transformed data $\mathcal{D}_s^w = Trans_w(\mathcal{D}_s)$, we generate pseudo-labels $\hat{y}^w = argmax(p^w)$ for weakly transformed sample $x^w \in \mathcal{D}_s^w$, where $p^w = \mathbf{C}(\mathbf{G}(x^w))$ is the output of classifier $\mathbf{C}$. FixMatch penalizes by assigning the pseudo-label $\hat{y}^w$ as the true label for strongly transformed sample $x^s \in \mathcal{D}_s^s$ with cross-entropy loss:

$$\mathcal{L}_{FM}(x) = -\mathbb{I}(\max(p^w) \geq \rho) \log p^s(\hat{y}^w), \tag{9}$$

where $p^s = \mathbf{C}(\mathbf{G}(x^s))$ and $\rho$ is the threshold to decide whether $\mathcal{L}_{FM}(x)$ is applied on sample $x$.

Further, with strongly and weakly transformed $\mathcal{D}_s^s$ and $\mathcal{D}_s^w$, we adopt our proposed label-free adversarial data augmentation (Figure 2 "Stage 1-2: Data Augmentation") to generate $\mathcal{D}_g^s$ and $\mathcal{D}_g^w$, respectively. With the generated $\mathcal{D}_g = \{\mathcal{D}_g^s, \mathcal{D}_g^w\}$, as shown in Figure 2 "Stage 2: Multi-Pair FixMatch", we further apply strong transformation on top of generated $\mathcal{D}_g^s$ and $\mathcal{D}_g^w$ to derive $\mathcal{D}_g^{ss}$ and $\mathcal{D}_g^{ws}$ as the input pairs. These two types of data will be paired with $\mathcal{D}_s^w$ to create two additional pairs for the MPFM loss:

$$\mathcal{L}_{MPFM}(x) = -\mathbb{I}(\max(p^w) \geq \rho)(\log p_g^{ss}(\hat{y}^w) + \log p_g^{ws}(\hat{y}^w))/2, \tag{10}$$

where $p_g^{ss} = \mathbf{C}(\mathbf{G}(x_g^{ss}))$, $x_g^{ss} \in \mathcal{D}_g^{ss}$ and $p_g^{ws} = \mathbf{C}(\mathbf{G}(x_g^{ws}))$, $x_g^{ws} \in \mathcal{D}_g^{ws}$. The objective for updating our model $(\mathbf{G}, \mathbf{C})$ in stage1-2 is:

$$\mathcal{L}_{S2} = \mathcal{L}_{ce} + \lambda \mathcal{L}_{FM} + \gamma \mathcal{L}_{MPFM}, \tag{11}$$

where $\lambda, \gamma$ are balancing hyper-parameters and empirically set as 1 and 0.5, respectively. $\mathcal{L}_{CE}$ is cross-entropy loss over labeled source domain data. Our method procedure is summarized in Algorithm 1.

---

**Algorithm 1** Our proposed Algorithm.

---

**Require:** $\mathcal{D}_s, \mathcal{D}_g, \mathcal{D}_g^w, \mathcal{D}_g^s$, initialized **G**, **F**, **C**.
**Ensure:** Learned **G** and classifier **C**
  1: **for** $t = 1$ to $T_P$ **do**                                                   $\triangleright T_P$ #iterations
  2:      Apply Eqn. 2 for Pre-train (Sec. 3.2)
  3:      **if** $\text{Mod}(t,Q)=0$ **then**                                    $\triangleright Q$ iteration interval
  4:         Apply Eqn. 8 for Data Augmentation (Sec. 3.3)
  5:      **end if**
  6: **end for**
  7: **for** $t = 1$ to $T_J$ **do**                                                   $\triangleright T_J$ #iterations
  8:      Group $x_g^s, x_g^w, x, x_l$
  9:      Apply *S-Aug* on $x_g^s, x_g^w, x$ and *W-Aug* on $x, x_l$
10:      Apply overall loss Eqn. 11 to train **G**, **C**
11: **end for**

---

# 4 Experiments

In this section, we evaluate the effectiveness of our method as the following: Firstly, we introduce the Experimental settings. Next, we compare our method to other methods mainly from single domain generalization and semi-supervised learning. Then, we provide an extensive ablative study investigating each of our proposed modules. Last, we present the latent feature space visualization.

## 4.1 Experimental Settings

### 4.1.1 Datasets

We leverage three commonly used datasets i.e., *PACS*, *OfficeHome*, and *DomainNet20*.

**(1) PACS** (Li et al., 2017) is a recent challenging domain adaptation/generalization benchmark which shows larger domain discrepancy. It consists of seven object categories from four domains, namely art paintings, cartoon, sketch, and photo. For this dataset, we evaluate on two SS-SDG settings: 15 labeled samples per class (total 105 labels) and 25 labeled samples per class (total 175 labels). **(2) OfficeHome** (Venkateswara et al., 2017) contains four domains (art, clipart, product, and real world) with 65 classes. This is one of the canonical domain adaptation/generalization benchmarks. We design two SS-SDG settings on this dataset: 10 labeled samples per class (total 650 labels) and 15 labeled samples per class (total 975 labels). **(3) DomainNet20** is a subset of DomainNet. We self-construct the setting as picking 4 domains (clipart, painting, real, and sketch) and 20 classes (0-baseball bat, 1-binoculars, 2-bracelet, 3-diving board, 4-goatee, 5-hamburger, 6-hurricane, 7-knee, 8-parachute, 9-pickup truck, 10-pillow, 11-pizza, 12-sandwich, 13-saw, 14-scorpion, 15-speedboat, 16-square, 17-swing set, 18-tent, 19-trumpet) out of the entire dataset. The reason why we create a subset of original DomainNet is that, we find that many categories are with limited samples. When we consider the settings of 15 or 20 samples per class, not all the categories defined in DomainNet are valid. Moreover, to indicate the same behavior trend of our trained models, we choose to trim the dataset to at the same scale of the other two datasets, OfficeHome and PACS. In this way, we select 4 domains and 20 categories to form DomainNet20. We adopt two SS-SDG settings: 15 labeled samples per class (total 300 labels) and 25 labeled samples per class (total 500 labels).

### 4.1.2 Baselines

We compare with two main streams of the state-of-the-art methods: (1) *Single domain generalization methods*, namely Adversarial Data Augmentation (ADA) (Volpi et al., 2018) and Maximum-Entropy Adversarial Data Augmentation (MEADA) (Zhao et al., 2020). (2) *Semi-supervised learning methods*, namely Entropy Minimization (ENT-MIN) (Yves Grandvalet, 2004) and FixMatch (Sohn et al., 2020). Besides, we also provide two kinds of baselines: (1) *Integration of Single Domain Generalization and Semi-Supervised Learning Techniques*: This approach combines single domain generalization and semi-supervised learning methods,

specifically ADA+FixMatch and MEADA+FixMatch. Here, we directly incorporate the semi-supervised learning method, i.e., FixMatch, with single domain adaptation methods, namely ADA and MEADA. (2) *Single Domain Generalization Methods with Pseudo Labels*: In this strategy, denoted as ADA+PL and MEADA+PL, we employ FixMatch to assign pseudo labels to unlabeled source data. Subsequently, ADA and MEADA are applied to the entire dataset, which includes both labeled and pseudo-labeled source data. Furthermore, we include another baseline, namely Empirical Risk Minimization (ERM) (Koltchinskii, 2011), and explore the potential upper bounds achievable when full labeled source domain data is available, both with ERM (supervised) and ADA (supervised).

### 4.1.3 Evaluation Metrics:

For each of the benchmarks, amongst all the domains defined, we iteratively take one domain as the source domain and test on all the rest domains. The mean average precision (mAP) is reported by averaging over 3 random splits of all the classes' average precision.

### 4.1.4 Implementation Details:

**For our framework:** at stage1 in Sec. 3.2 and 3.3, We adopt an ImageNet-pretrained ResNet18 as the feature extractor **G** and a 2-layer MLP head (hidden layer 512-d, with ReLU) as projection head **F**. We set memory bank size to 1600 and batch size to 32 in whole training process. At the model pre-train stage in Sec. 3.2, we use SGD optimizer with learning rate 0.0005, weight decay 0.0005 and momentum 0.9, and train for 1500 iterations. At data augmentation stage in Sec. 3.3, we adopt SGD optimizer with learning rate 50.0 and 15 iterations to maximize Equation 7. The data augmentation is involved only once at the 200-th iteration during model pre-train. At "Stage 2" in Sec. 3.4, we adopt SGD with learning rate 0.001, weight decay 0.0005, momentum 0.9, batch size 32 and train for 8500 iterations.

**For baselines:** the ADA and MEADA perform adversarial data augmentation with the labeled source data. ADA+FixMatch and MEADA+FixMatch jointly apply adversarial data augmentation on labeled source data and utilize FixMatch on unlabeld source data. ADA+PL and MEADA+PL firstly employ FixMatch to train a model and assign pseudo-labels to the unlabeled source data, and then apply adversarial data augmentation to augment the entire source domain, which includes both the source data with ground truth labels and the source data with pseudo labels.

## 4.2 Performance Evaluation

### 4.2.1 Results on PACS

Quantitative comparison is shown in Table 1. We group the methods on top three rows compared to ours on the last row. On the first row, notice that ERM (supervised) and ADA (supervised) use all the dataset labels for supervised learning hence serves as the upper bound, where it is neither in the 15 label per class setting nor in the 25 per class setting. We equip ERM with our semi-supervised domain generalization setting for the fair comparison. On the second row, there are two semi-supervised learning methods. One is canonical, the entropy minimization method, the other is recent top method FixMatch. On the third row, we present two cutting-edge domain generalization methods: Adversarial Data Augmentation (ADA) and Maximum-Entropy Adversarial Data Augmentation (MEADA). In the fourth row, we introduce ADA and MEADA as two potential frameworks and combine them with FixMatch, resulting in ADA+FixMatch and MEADA+FixMatch. Moving to the fifth row, we utilize FixMatch on the source domain to assign pseudo-labels for unlabeled source data. Subsequently, we apply ADA and MEADA to the source domain, considering both labeled and pseudo-labeled source data in ADA+PL and MEADA+PL, respectively. On 15 instance per class setting, we observe that our method consistently outperforms all the compared methods with significant margin. For example, we get 2.43% better than FixMatch and 14.48% better than MEADA in terms of "Avg". By checking the 25 labels per class setting, we see 2.4% and 15.92% performance gains on "Painting" compared with FixMatch and MEADA, respectively. Further, surprisingly our "Avg." number is approaching the upper bound ERM (supervised) which utilizes all the label information from the dataset within less than 4% gap, whereas our method only utilizes less than 5% (15 per class) or 8% (25 per class) of all the labels. Furthermore, we have observed that simply combining single domain generalization methods

Table 1: mAP(%) on PACS (averaged over 3 random splits). Named in column is source domain for training. Rest domains are the testing unseen domains and precision for each testing domain is averaged and reported. (A: Art painting, C: Cartoon, P: Photo, S: Sketch)

| Method | labels: 105 (15 per class) | | | | | labels: 175 (25 per class) | | | | |
|---|---|---|---|---|---|---|---|---|---|---|
| | A | C | P | S | Avg. | A | C | P | S | Avg. |
| ERM (supervised) | 70.9 | 76.5 | 43.3 | 53.1 | 60.7 | 70.9 | 76.5 | 43.3 | 53.1 | 60.7 |
| ADA (supervised) | 72.43 | 71.97 | 44.63 | 45.73 | 58.70 | 72.43 | 71.97 | 44.63 | 45.73 | 58.70 |
| ERM | 49.45 | 50.84 | 30.36 | 25.25 | 38.96 | 52.51 | 54.05 | 30.76 | 23.61 | 40.23 |
| ENT-MIN | 54.15 | 55.61 | 38.08 | 24.95 | 43.20 | 56.10 | 59.65 | 36.99 | 26.92 | 44.91 |
| FixMatch | 57.67 | 69.13 | 45.46 | 42.17 | 53.61 | 63.70 | 68.27 | 45.79 | 42.86 | 55.16 |
| ADA | 47.87 | 50.71 | 30.31 | 30.46 | 40.08 | 51.71 | 53.42 | 30.33 | 26.74 | 40.55 |
| MEADA | 48.79 | 52.81 | 34.23 | 30.42 | 41.56 | 51.97 | 54.54 | 32.27 | 28.19 | 41.74 |
| ADA+FixMatch | 60.02 | 69.44 | 45.36 | 39.24 | 53.51 | 59.73 | 68.97 | 44.31 | 40.58 | 53.40 |
| MEADA+FixMatch | 59.99 | 69.65 | 46.41 | 40.55 | 54.15 | 61.12 | 70.10 | 47.41 | 40.81 | 54.86 |
| ADA+PL | 59.31 | 65.69 | 43.23 | 39.31 | 51.89 | 59.63 | 66.57 | 44.43 | 40.21 | 52.71 |
| MEADA+PL | 62.17 | 61.92 | 41.66 | 37.16 | 50.72 | 59.88 | 65.92 | 44.59 | 41.44 | 52.96 |
| Ours | **60.26** | **69.91** | **47.30** | **46.70** | **56.04** | **65.58** | **70.11** | **48.19** | **47.95** | **57.96** |

Table 2: mAP(%) on OfficeHome (averaged over 3 random splits). Named in column is source domain for training. Rest domains are testing unseen domains and precision for each testing domain is averaged and reported. (A: Art, C: Clipart, P: Product, R: Real_world)

| Method | labels: 650 (10 per class) | | | | | labels: 975 (15 per class) | | | | |
|---|---|---|---|---|---|---|---|---|---|---|
| | A | C | P | R | Avg. | A | C | P | R | Avg. |
| ERM (supervised) | 51.27 | 49.18 | 44.12 | 56.86 | 50.36 | 51.27 | 49.18 | 44.12 | 56.86 | 50.36 |
| ADA (supervised) | 53.24 | 50.02 | 45.76 | 57.36 | 51.60 | 53.24 | 50.02 | 45.76 | 57.36 | 51.60 |
| ERM | 41.09 | 35.57 | 36.09 | 44.15 | 39.23 | 44.94 | 38.99 | 37.84 | 47.85 | 42.40 |
| ENT-MIN | 40.74 | 37.10 | 38.69 | 46.98 | 40.88 | 45.43 | 40.93 | 39.96 | 51.09 | 44.35 |
| FixMatch | 40.52 | 36.53 | 39.19 | 47.73 | 40.99 | 44.70 | 41.72 | 40.21 | 52.04 | 44.67 |
| ADA | 40.75 | 35.63 | 35.83 | 43.76 | 38.99 | 44.67 | 38.97 | 37.16 | 47.43 | 42.05 |
| MEADA | 40.74 | 35.80 | 35.86 | 44.15 | 39.14 | 44.97 | 39.41 | 37.14 | 47.71 | 42.31 |
| ADA+FixMatch | 41.41 | 38.82 | 40.53 | 47.62 | 42.10 | 41.26 | 41.56 | 40.60 | 51.48 | 43.73 |
| MEADA+FixMatch | 41.79 | 38.72 | 40.19 | 47.73 | 42.11 | 44.70 | 41.72 | 41.91 | 53.04 | 45.34 |
| ADA+PL | 38.57 | 37.00 | 36.35 | 45.13 | 39.26 | 42.68 | 39.05 | 38.77 | 48.84 | 42.34 |
| MEADA+PL | 39.17 | 37.36 | 37.61 | 45.43 | 39.89 | 42.13 | 40.30 | 39.42 | 48.01 | 42.47 |
| Ours | **43.55** | **40.65** | **40.54** | **49.13** | **43.47** | **47.67** | **43.39** | **42.49** | **53.46** | **46.75** |

with semi-supervised learning methods, such as ADA+FixMatch and MEADA+FixMatch, does not yield comparable results to our approach. Additionally, the baselines, ADA+PL and MEADA+PL, encounter challenges due to the unsatisfied quality of the pseudo-labels, resulting in the performance that also falls short of achieving results comparable to our method.

### 4.2.2 Results on OfficeHome

Evaluation on OfficeHome is shown in Table 2. Similarly as PACS, we organize the experiment in the same layout, where the same five groups of the compared methods are listed on the top three rows with EMR (supervised) and ADA (supervised) as the upper bound. In this evaluation, again we find that our method consistently and significantly outperforms all the compared methods on both of the two settings in Table 2. On 10 instance per class setting, we observe that our method consistently outperforms all the compared methods with significant margin, i.e., 2.8% better than second best on "Art" as source domain and 3.55% better than second best on "Clipart". The same trend is observed on 15 instances per class setting on the right column. Besides, as more instances per class is used in training, we see that the performance on the right column is generally higher than the left column setting as expected. Another interesting point is while

Table 3: mAP(%) on DomainNet20 (averaged over 3 random splits). Named in column is source domain for training. Rest domains are the testing unseen domains and precision for each testing domain is averaged and reported. (C: Clipart, P: Painting, R: Real, S: Sketch)

| Method | labels: 300 (15 per class) | | | | | labels: 500 (25 per class) | | | | |
|---|---|---|---|---|---|---|---|---|---|---|
| | C | P | R | S | Avg. | C | P | R | S | Avg. |
| ERM (supervised) | 59.57 | 64.74 | 55.57 | 55.82 | 58.93 | 59.57 | 64.74 | 55.57 | 55.82 | 58.93 |
| ADA (supervised) | 60.22 | 65.69 | 56.17 | 56.07 | 59.54 | 60.22 | 65.69 | 56.17 | 56.07 | 59.54 |
| ERM | 44.87 | 52.27 | 39.38 | 42.34 | 44.72 | 48.81 | 54.21 | 44.56 | 47.47 | 48.76 |
| ENT-MIN | 45.22 | 53.78 | 46.84 | 42.36 | 47.05 | 50.93 | 53.14 | 49.59 | 49.73 | 50.83 |
| FixMatch | 51.45 | 55.27 | 50.17 | 49.56 | 51.61 | 54.23 | 57.42 | 50.35 | 53.26 | 53.72 |
| ADA | 45.28 | 52.11 | 39.69 | 43.10 | 45.05 | 48.93 | 53.68 | 44.82 | 48.74 | 49.04 |
| MEADA | 46.48 | 52.55 | 40.93 | 43.37 | 45.83 | 49.74 | 53.50 | 45.08 | 49.50 | 49.46 |
| ADA+FixMatch | 51.47 | 52.25 | 47.91 | 50.58 | 50.55 | 54.25 | 54.85 | 51.04 | 52.61 | 53.19 |
| MEADA+FixMatch | 52.13 | 49.74 | 49.88 | 48.10 | 49.96 | 54.28 | 51.90 | 53.15 | 53.12 | 53.11 |
| ADA+PL | 43.79 | 51.06 | 39.27 | 45.24 | 44.84 | 48.94 | 53.07 | 45.60 | 49.85 | 49.37 |
| MEADA+PL | 46.35 | 53.03 | 39.88 | 43.57 | 45.71 | 49.11 | 52.98 | 45.78 | 51.02 | 49.72 |
| Ours | **53.31** | **57.52** | **52.61** | **52.91** | **54.09** | **56.21** | **59.35** | **53.38** | **56.18** | **56.28** |

our performance to upper bound gap is less than 7%, our setting only uses less than 18% labels of OfficeHome (10 instances per class) and 26.71% (15 instances per class). Furthermore, we still find that our label-free adversarial data augmentation based framework consistently and significantly outperforms ADA, MEADA, and their variant combinations, with more than 10.0% advantage under different settings.

### 4.2.3   Results on DomainNet20

The original DomainNet (Peng et al., 2019) is relatively a large scale domain adaptation benchmark with 6 domains and overall 345 categories. Since our setting is semi-supervised, where we only utilize a few labeled data, e.g., we constrain only 15 or 25 samples per class presenting their labels. In this way, for those categories that are with less than 15 images, we discard them from the training data. Meanwhile, to provide a similar-scale evaluation to other benchmarks such as OfficeHome and PACS, we pick 4 out of 6 domains and 20 out of the 345 categories which contain much more samples in each category to form our training and testing data for demonstration. In Table 3, our method's performance across the 300-label and 500-label settings show clear advantage over the other methods. Compared to the most competitive opponent FixMatch, our method surpasses by 2.47% on 300-label "Avg." and 2.56% on 500-label "Avg.". Compared to the representative SDG methods, i.e., MEADA, we see 8.26% performance gain on 300-label "Avg." and 6.82% performance gain 500-label "Avg.". In this dataset, we observe that the ERM baseline performs at the same level as those single domain generalization methods, partially suggesting that this dataset is more challenging as label information other than domain discrepancy becomes more critical, where the self-supervised methods can benefit more. Notice that our method "Avg" presents close performance to the upper bound ERM (supervised) and ADA (supervised) with less than 5% accuracy gap, but using only 7.7% (15 per class) or 12.96% (25 per class) of the overall labels, which further indicates the effectiveness of our self-supervised adversarial data augmentation design.

### 4.3   Ablation Study

We ablate our proposed core components in Table 4. We clearly decompose the loss terms and abbreviate the simple baseline (ERM) as "BL" and the strong baseline FixMatch as "FM". $\mathcal{L}_{CL}$ refers to our pre-training stage 1-1 and data augmentation stage 1-2. $\mathcal{L}_{MPFM}$ refers to that, in stage 1-2, we further conduct LFADA on weakly and strongly transformed data to generate data for the multi-pair loss. Compared to the naive ERM "BL", the strong baseline "FM" achieves much better performance. Our intention was not to show the progressive improvement on accuracy from "BL" to "FM" and then to "Ours". Indeed, we aim to demonstrate that proposed $\mathcal{L}_{MPFM}$ and $\mathcal{L}_{CL}$ can orthogonally improve the performance of baselines, e.g., comparing "BL" to "BL+S1", and "FM" to "FM+S1", where a clear boost of 1.8% and 0.8% is observed,

Table 4: Ablation study on PACS with 15 per class setting. Named in column is source domain. Rest domains are the testing unseen domains. (A: Art painting, C: Cartoon, P: Photo, S: Sketch)

| PACS | $\mathcal{L}_{ce}$ | $\mathcal{L}_{FM}$ | $\mathcal{L}_{CL}$ | $\mathcal{L}_{MPFM}$ | A | C | P | S | Avg. |
|---|---|---|---|---|---|---|---|---|---|
| BL | ✓ | | | | 49.45 | 50.84 | 30.36 | 25.25 | 38.96 |
| BL+S1 | ✓ | | ✓ | | 50.60 | 52.40 | 32.75 | 27.78 | 40.88 |
| FM | ✓ | ✓ | | | 57.67 | 69.13 | 45.46 | 42.17 | 53.61 |
| FM+S1 | ✓ | ✓ | ✓ | | 59.39 | 69.61 | 45.67 | 43.02 | 54.42 |
| Ours | ✓ | ✓ | ✓ | ✓ | **60.26** | **69.91** | **47.30** | **46.70** | **56.04** |

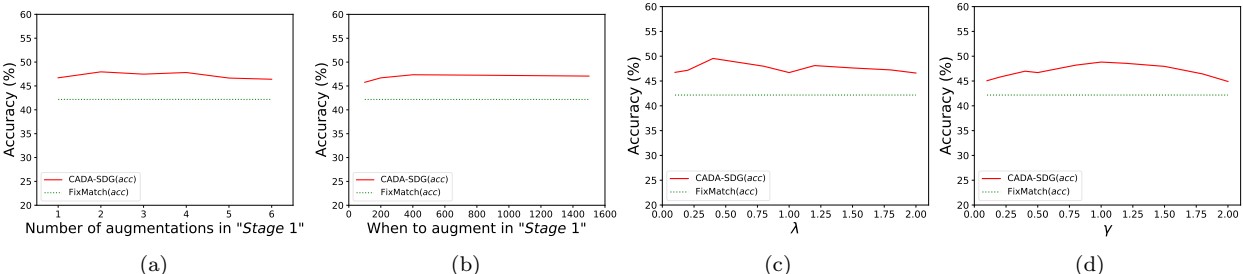

(a)       (b)       (c)       (d)

Figure 3: (a) Accuracy versus different number of augmentations in "Stage 1". (b) Accuracy versus when to augment in "Stage 1". (c) Accuracy versus different values of $\lambda$. (d) Accuracy versus different values of $\gamma$. Here, the Sketch, comes from $PACS$ dataset, is used as source domain

respectively. Furthermore, by comparing "Ours" to "FM+S1" and "FM', we achieve another 1.6% and 2.4% accuracy gain. Each of the incremental combination demonstrates the effectiveness of the components, such as the label-free adversarial data augmentation and the Multi-Pair FixMatch, suggesting the LFADA is advantageous to handle semi-supervised single domain generalization (SS-SDG) problems.

### 4.4 Number of Augmentations in "Stage 1"

We study the effect of the hyper-parameter: the number of augmentations in "Stage 1", on $PACS$ dataset where Sketch is adopted as source domain. We plot the accuracy curve under different augmentation times. As shown in Fig. 3 (a), we find that the accuracy reaches the best value when augmenting twice and the accuracy is slowly getting worse with further increased the number of augmentation.

### 4.5 When to Do Augmentation in "Stage 1"

We explore the influence of when to augment in "Stage 1" on $PACS$ dataset where Sketch is used as source domain. The experimental results are reported in Fig. 3 (b). Here, the maximum iteration in "Stage 1" is set to 1500. It can be observed that the performance of our method is improving with augmentation being executed later in "Stage 1". Because a more well pre-trained model in "Stage 1" benefits more on our data augmentation process. After 800 iterations, the choice of different iterations for data augmentation is very close, demonstrating that the pre-trained model in "Stage 1" after 800 iterations is already converged.

### 4.6 Hyper-parameters Sensitivity Study

We discuss the sensitivity of hyper-parameters $\lambda$ and $\gamma$ in Eq. equation 11 by evaluating them on $PACS$ dataset where Sketch is selected as source domain. The value of hyper-parameter $\lambda$ is selected from {0.1, 0.2, 0.4, 0.8, 1.0, 1.2, 1.5, 1.8, 2.0} and the value of hyper-parameter $\gamma$ is fixed to 0.5. Fig. 3 (c) shows that our method is relatively stable in the range [0.1, 2.0] and gets better performance in the range [0.25, 0.75]. The value of hyper-parameter $\gamma$ is selected from {0.1, 0.2, 0.4, 0.5, 0.8, 1.0, 1.2, 1.5, 1.8, 2.0} and the value

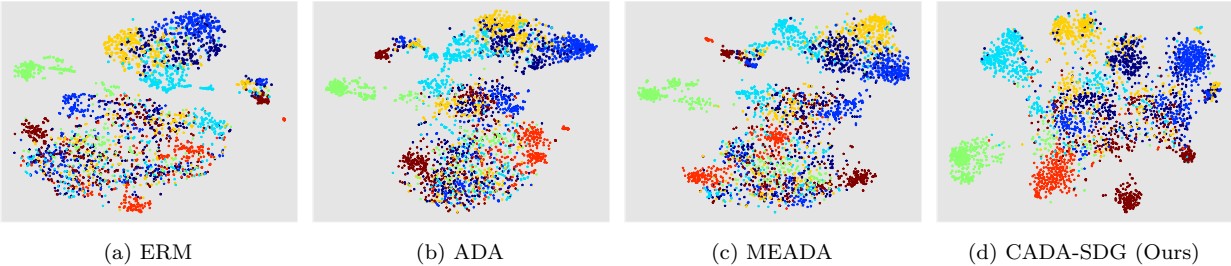

(a) ERM       (b) ADA       (c) MEADA       (d) CADA-SDG (Ours)

Figure 4: The t-SNE visualization of feature space with baselines, i.e., ERM, ADA, and MEADA, and our method. Same color indicates the same category. Circle indicates source domain. Star indicates unseen target domains. Best viewed in color and zoomed in.

of $\lambda$ is fixed to 1.0. As shown in Fig. 3 (d), the classification accuracy is relatively stable in the range [0.75, 1.5]. When $\gamma$ is out of the range [0.75, 1.5], the performance is slightly degraded.

### 4.7 Visualization

To give an intuitive understanding of our proposed method, we visualize feature space extracted by models trained with our method and the baseline methods, i.e., ERM, ADA, and MEADA, on PACS dataset where Sketch is the target domain. As shown in Figure 4, It appears that our model yields clear better separation of different categories. Meanwhile, we observe that for different shapes in the same color, with our method, the data points tend to cluster closer, while other methods leave those same classes but different domain data points separated. The closer clustering indicates better alignment of source domain to target domains. Our method presents clear lower domain gap than baselines, i.e., ERM, ADA, and MEADA, demonstrating that our method indeed continuously generalizes towards the unseen target domains.

## 5 Conclusion

In this work, we propose to solve a new problem under a realistic setting, namely the semi-supervised single domain generalization, where number of domains for generalization is single and with only very limited label information of training data. We leverage expertise from self-supervised learning and propose a multi-pair FixMatch loss to mitigate the lack of label issue. Further, we newly introduce a label-free adversarial data augmentation to enrich the source domain distribution under insufficient label information scenario. Our augmentation is significantly different from traditional adversarial ones that heavily rely on label information. Extensive study across three challenging domain generalization benchmarks demonstrates our method's advantage, not only surpassing state-of-the-art methods, but also approaching the fully supervised upper bounds.

**Acknowledgement** The work of R. Zhu and S. Li is in part supported by the the U.S. Army Research Office Award under Grant Number W911NF-21-1-0109 and the National Science Foundation under Grant IIS-2316306.

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

Table 5: std(%) on PACS (averaged over 3 random splits). Named in column is source domain for training. Rest domains are the testing unseen domains and precision for each testing domain is averaged and reported. (A: Art painting, C: Cartoon, P: Photo, S: Sketch)

| Method | labels: 105 (15 per class) | | | | | labels: 175 (25 per class) | | | | |
|---|---|---|---|---|---|---|---|---|---|---|
| | A | C | P | S | Avg. | A | C | P | S | Avg. |
| ERM (supervised) | 0.26 | 0.33 | 0.36 | 0.42 | 0.34 | 0.23 | 0.34 | 0.26 | 0.31 | 0.28 |
| ADA (supervised) | 0.31 | 0.34 | 0.39 | 0.45 | 0.35 | 0.23 | 0.23 | 0.31 | 0.24 | 0.25 |
| ERM | 0.46 | 0.54 | 0.49 | 0.45 | 0.47 | 0.42 | 0.38 | 0.56 | 0.49 | 0.46 |
| ENT-MIN | 0.36 | 0.35 | 0.46 | 0.39 | 0.38 | 0.33 | 0.48 | 0.46 | 0.45 | 0.43 |
| FixMatch | 0.31 | 0.38 | 0.33 | 0.35 | 0.36 | 0.33 | 0.29 | 0.28 | 0.34 | 0.31 |
| ADA | 0.41 | 0.44 | 0.39 | 0.40 | 0.42 | 0.36 | 0.32 | 0.46 | 0.42 | 0.41 |
| MEADA | 0.36 | 0.41 | 0.49 | 0.35 | 0.41 | 0.32 | 0.39 | 0.36 | 0.39 | 0.38 |
| ADA+FixMatch | 0.38 | 0.45 | 0.39 | 0.35 | 0.38 | 0.32 | 0.41 | 0.36 | 0.29 | 0.36 |
| MEADA+FixMatch | 0.35 | 0.39 | 0.33 | 0.41 | 0.37 | 0.37 | 0.29 | 0.36 | 0.39 | 0.37 |
| ADA+PL | 0.36 | 0.38 | 0.36 | 0.41 | 0.38 | 0.36 | 0.28 | 0.36 | 0.35 | 0.35 |
| MEADA+PL | 0.31 | 0.33 | 0.38 | 0.37 | 0.35 | 0.33 | 0.31 | 0.33 | 0.32 | 0.32 |
| Ours | 0.31 | 0.35 | 0.34 | 0.37 | 0.35 | 0.31 | 0.30 | 0.27 | 0.28 | 0.29 |

Table 6: std(%) on OfficeHome (averaged over 3 random splits). Named in column is source domain for training. Rest domains are testing unseen domains and precision for each testing domain is averaged and reported. (A: Art, C: Clipart, P: Product, R: Real_world)

| Method | labels: 650 (10 per class) | | | | | labels: 975 (15 per class) | | | | |
|---|---|---|---|---|---|---|---|---|---|---|
| | A | C | P | R | Avg. | A | C | P | R | Avg. |
| ERM (supervised) | 0.30 | 0.35 | 0.27 | 0.32 | 0.32 | 0.24 | 0.26 | 0.27 | 0.27 | 0.25 |
| ADA (supervised) | 0.38 | 0.36 | 0.34 | 0.34 | 0.36 | 0.27 | 0.24 | 0.25 | 0.28 | 0.26 |
| ERM | 0.35 | 0.43 | 0.44 | 0.38 | 0.41 | 0.38 | 0.34 | 0.36 | 0.39 | 0.37 |
| ENT-MIN | 0.37 | 0.43 | 0.41 | 0.35 | 0.39 | 0.33 | 0.37 | 0.40 | 0.33 | 0.35 |
| FixMatch | 0.38 | 0.39 | 0.32 | 0.31 | 0.35 | 0.29 | 0.21 | 0.28 | 0.25 | 0.26 |
| ADA | 0.37 | 0.43 | 0.39 | 0.38 | 0.39 | 0.35 | 0.27 | 0.38 | 0.36 | 0.34 |
| MEADA | 0.41 | 0.43 | 0.38 | 0.37 | 0.40 | 0.35 | 0.39 | 0.31 | 0.37 | 0.36 |
| ADA+FixMatch | 0.41 | 0.35 | 0.37 | 0.35 | 0.38 | 0.31 | 0.27 | 0.33 | 0.29 | 0.30 |
| MEADA+FixMatch | 0.34 | 0.40 | 0.38 | 0.31 | 0.36 | 0.27 | 0.28 | 0.29 | 0.35 | 0.29 |
| ADA+PL | 0.37 | 0.41 | 0.39 | 0.44 | 0.41 | 0.37 | 0.39 | 0.34 | 0.35 | 0.36 |
| MEADA+PL | 0.38 | 0.44 | 0.37 | 0.41 | 0.40 | 0.35 | 0.38 | 0.32 | 0.33 | 0.35 |
| Ours | 0.33 | 0.34 | 0.29 | 0.31 | 0.32 | 0.25 | 0.24 | 0.26 | 0.25 | 0.25 |

# A  Appendix

## A.1  Standard Deviation of Results

The standard deviation of the experimental results on PACS, OfficeHome, and DomainNet20 are show in Tables 5, 6, 7, respectively.

## A.2  Visualization of Generated Data

We visualize the strongly transformed generated data, e.g., $\mathcal{D}_g^{ss}$, used in Eq.(10). As shown in Figure 5 (a), besides the overlap of generated and original data, there is a distinct portion of non-overlapping generated data represented by the green dots in the figure. Furthermore, the visualization demonstrates that the original and generated samples are notably distant from each other, as indicated by the color connections. This observation effectively confirms the enrichment of data diversity through our approach.

Table 7: std(%) on DomainNet20 (averaged over 3 random splits). Named in column is source domain for training. Rest domains are the testing unseen domains and precision for each testing domain is averaged and reported. (C: Clipart, P: Painting, R: Real, S: Sketch)

| Method | labels: 300 (15 per class) | | | | | labels: 500 (25 per class) | | | | |
|---|---|---|---|---|---|---|---|---|---|---|
| | C | P | R | S | Avg. | C | P | R | S | Avg. |
| ERM (supervised) | 0.51 | 0.48 | 0.53 | 0.57 | 0.52 | 0.43 | 0.44 | 0.48 | 0.47 | 0.45 |
| ADA (supervised) | 0.58 | 0.46 | 0.56 | 0.54 | 0.53 | 0.47 | 0.46 | 0.51 | 0.41 | 0.46 |
| ERM | 0.61 | 0.55 | 0.59 | 0.61 | 0.59 | 0.58 | 0.48 | 0.53 | 0.54 | 0.52 |
| ENT-MIN | 0.49 | 0.63 | 0.51 | 0.57 | 0.55 | 0.42 | 0.55 | 0.39 | 0.49 | 0.46 |
| FixMatch | 0.38 | 0.43 | 0.52 | 0.49 | 0.45 | 0.35 | 0.39 | 0.28 | 0.33 | 0.34 |
| ADA | 0.63 | 0.51 | 0.55 | 0.58 | 0.57 | 0.51 | 0.43 | 0.49 | 0.50 | 0.48 |
| MEADA | 0.59 | 0.57 | 0.61 | 0.55 | 0.58 | 0.48 | 0.50 | 0.42 | 0.47 | 0.47 |
| ADA+FixMatch | 0.45 | 0.51 | 0.48 | 0.47 | 0.58 | 0.48 | 0.45 | 0.41 | 0.39 | 0.43 |
| MEADA+FixMatch | 0.39 | 0.61 | 0.53 | 0.47 | 0.51 | 0.32 | 0.45 | 0.49 | 0.43 | 0.42 |
| ADA+PL | 0.57 | 0.58 | 0.49 | 0.41 | 0.51 | 0.48 | 0.42 | 0.43 | 0.44 | 0.44 |
| MEADA+PL | 0.60 | 0.54 | 0.57 | 0.55 | 0.56 | 0.38 | 0.38 | 0.45 | 0.47 | 0.42 |
| Ours | 0.45 | 0.41 | 0.44 | 0.34 | 0.41 | 0.32 | 0.37 | 0.29 | 0.28 | 0.32 |

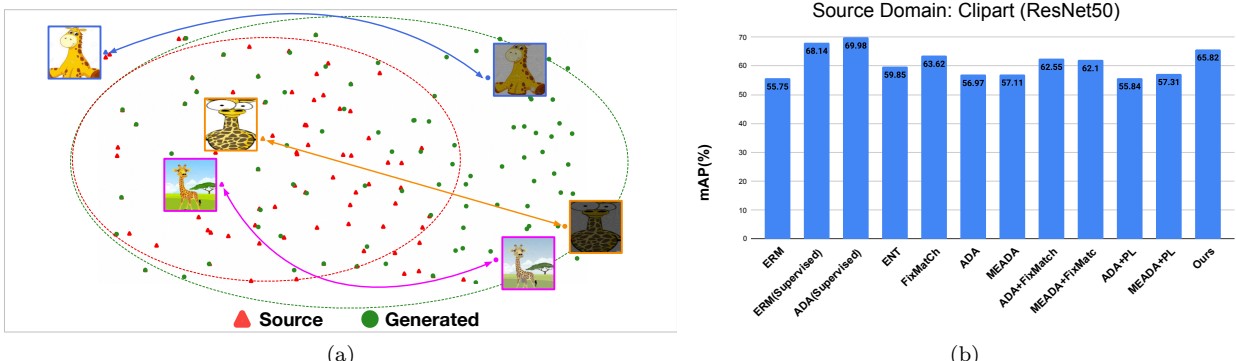

(a)                                                    (b)

Figure 5: (a) Visualization of features of source and generated samples ("Giraffe" category in Cartoon domain of PACS). (b) mAP (%) on DomainNet20 with Clipart as the source domain. ResNet50 is selected as the feature extractor.

## A.3 Results on DomainNet20 with ResNet50

As illustrated in Figure 5 (b), we chose the Clipart domain as our source domain and employed ResNet50 as the feature extractor during training. The remaining domains, namely Painting, Real, and Sketch, were designated as unseen testing domains. Our results demonstrate that our method achieves comparable performance to two upper bounds, namely ERM (supervised) and ADA (supervised). Additionally, it holds a significant advantage over the best baseline, FixMatch, with an improvement of more than 2%. This outcome strongly validates the effectiveness of our approach.

## A.4 Ablation Study on $D_g^{ss}$ and $D_g^{ws}$

We conducted a deeper investigation into the dataset $D_g^{ss}$ and $D_g^{ws}$, which is applied with strong transformation, as described in Equation (10). Specifically, we compared our proposed method with a variant that did not apply strong transformations to the combined dataset $D_g = D_g^s, D_g^w$. The results, as displayed in Table 8, clearly demonstrate that applying strong transformations to $D_g$ consistently leads to performance improvements.

Table 8: Ablation study on PACS with 15 per class setting. Named in column is source domain. Rest domains are the testing unseen domains. (A: Art painting, C: Cartoon, P: Photo, S: Sketch)

| PACS | Using $D_g^{ss}$ and $D_g^{ws}$? | A | C | P | S | Avg. |
|------|------|------|------|------|------|------|
| Ours | | 59.76 | 68.45 | 46.71 | 45.98 | 55.23 |
| Ours | ✓ | **60.26** | **69.91** | **47.30** | **46.70** | **56.04** |

