# OpenReview forum: "Semi-Supervised Single Domain Generalization with Label-Free Adversarial Data Augmentation"
_TMLR — Accepted by TMLR_

### Review · Reviewer_j6st · 2023-07-13

**Summary Of Contributions:**

This paper introduces a novel task: semi-supervised single domain generalization, specifically tailored to situations that involve a single source domain with only partial labeling, yet need to generalize to multiple unseen target domains. The authors put forth a label-free adversarial data augmentation strategy, designed to amplify the diversity of a single source domain in the absence of label information. Additionally, they propose a Multi-pair FixMatch regularization, which capitalizes on the diversified data to further refine the generalized classifier for unseen target domains. Experimental findings substantiate the superior performance of the proposed method when compared to SOTA baselines.

**Audience:**

Yes

**Broader Impact Concerns:**

My concern is that the appearance of fundation models, is it necessary to consider DG again?

**Claims And Evidence:**

Yes

**Requested Changes:**

Please address the issues in weaknesses.

**Strengths And Weaknesses:**

Strengths:
1) the problem setting is novel to current DG community.
2) The authors further suggest a self-supervised model pretraining method employing contrastive learning, aimed at enhancing the quality of representations. Additionally, they introduce an adversarial data augmentation strategy designed to increase the diversity within the source domain.
3) The authors adeptly adapt the FixMatch method into a Multi-Pair FixMatch format, specifically tailored to address the SS-SDG problem. The empirical results underscore the efficacy of the proposed framework.
4) SOTA experimental performance!

Weaknesses:
1) The thechniques used is not novel. It seems the work is a combination. But it is ok to me.
2) Do you think semi-supervised source domain is really necessary? I mean it will impact the methods and experiments essentially?
3) Missing references: please cite the lasest paper related to DG, such as Moderately Distributional Exploration for Domain Generalization, Domain Generalization using Causal Matching

---

> ### Author Response · Authors · 2023-09-06
> **Response to Reviewer j6st**
>
> **Q1: Techniques contributions**
>
> **Response:** We agree with the reviewer that our overall framework is indeed a combination of several techniques. However, our contribution does not lie on the framework, but rather, each of the known technical modules with ‘’novel’’ improvements. For example, unlike well-known FixMatch, our approach employs a “semi-supervised loss”-driven auto-augmentation. Moreover, as indicated by our experiments, the simple combination of the known techniques actually show inferior performance compared to our proposed combination of newly designed modules.
>
> **Q2: Necessity and Impact of Semi-Supervised Source Domain**
>
> **Response:** We argue that SS-SDG is a new and yet important setting. We take a practical and widely applied scenario, face recognition, for example. Considering the initial version of a face recognition model is trained using offline labeled data. Under the real surveillance situation, the new users do not have the “enrollment”, where the newly captured data is unlabeled. We know that the offline training data as source data is fixed. However, the target domains are variant. It can be both indoor and outdoor, where the offline data v.s. newly captured data is labeled v.s. unlabeled. In this way, we have both labeled single source domain data and multiple unlabeled target domains’ data. Our newly proposed SS-SDG can exactly deal with this situation.
>
> **Q3: Missing latest paper related to DG**
>
> **Response:** Thank you for the valuable suggestion. We have incorporated the latest Domain Generalization (DG) papers into our related work section.
>
>
> **Q4: Necessity of Reconsidering Domain Generalization (DG) due to Foundation Models**
>
> **Response:** We appreciate the reviewer's question regarding the necessity of considering Domain Generalization (DG) in the presence of foundation models. To provide clarity on this matter, we would like to emphasize the following points:
> 1. Our problem arises in situations where data is not abundant, and labeled data is very limited. This scenario is practical and widely prevalent in real-world applications. In such cases, the data is often insufficient to support the direct training of foundation models. We are concerned that using foundation models in these circumstances might lead to suboptimal performance.
> 2. Despite the superior generalization capabilities of foundation models like CLIP, they often struggle to effectively tackle the challenges posed by domain gaps. Recently, numerous methods grounded in CLIP-based domain adaptation [1,2,3] and domain generalization [4, 5] have emerged with the aim of enhancing CLIP's generalization performance in the presence of domain gaps.
>
>
> [1] Chunjiang, Ge et al. (2022). “Domain adaptation via prompt learning.” In arXiv.
>
> [2] Giacomo, Zara et al. (2023). “AutoLabel: CLIP-based framework for open-set video domain adaptation.” In CVPR.
>
> [3] Yang, Shu et al. (2023). “CLIPood: Generalizing CLIP to out-of-distributions.” In ICML.
>
> [4] Vidit, Vidit et al. (2023). “CLIP the gap: A single domain generalization for object detection.” In CVPR.
>
> [5] Shirsha, Bose et al. (2023). “StyLIP: Multi-scale style-conditioned prompt learning for CLIP-based domain generalization.” In arXiv.

---

> > ### Comment · Reviewer_j6st · 2023-09-25
> > **To rebuttal**
> >
> > After reviewing the responses, all my concerns have been addressed.

---

> > > ### Comment · Action_Editors · 2023-09-25
> > >
> > > You need to post an "Official Recommendation". Thanks!

---

### Review · Reviewer_Ycg7 · 2023-07-25

**Summary Of Contributions:**

This paper considered out-of-distribution (or domain generalization) in the context of semi-supervised learning. Specifically, this paper considered two stages approaches. In the first stage, it contains (1) pre-training of the deep network through contrastive learning; (2) data-augmentation based adversarial sample generation; (3) confidence pseudo-label based approach.



**Audience:**

Yes

**Claims And Evidence:**

No

**Requested Changes:**

In general, I would think a significant revision on the paper's structure and presentation could improve the paper. Thus authors are suggested to check my detailed comments in the review.

*Actually, the claim and evidence are partially done. Since there is no such an option, I would currently recommend No. (This does not imply no claim or evidence). *

**Strengths And Weaknesses:**

### Summary

This paper considered domain generalization problems under the limited observed labelled dataset and unsupervised dataset. The problem setting sounds interesting and compelling. Simultaneously, this reviewer currently has several concerns, where the current recommendation is **borderline positive**.

### Would some individuals in TMLR's audience be interested in the findings of this paper?

Sure. This paper considers a relative novel setting in out-of-distribution generation.


### Are the claims made in the submission supported by accurate, convincing and clear evidence?
In general, the proposed method seems a bit complicated by building up different components. Upon reflection, most building components are believed to be reasonable. Simultaneously, it would be great to have discussion on the following points


- Why do we consider Wasserstein distance based adversarial samples? It could be possible to use other forms of attack?
- In Equation (9), how to set the threshold? Is this threshold important in your experiments?
- In Equation (10), is this related to prediction entropy regularization? Or to minimize the cross entropy loss?
- Could you explain why using the data memory in eq (2)? I could not find the rationale.
- It is suggested to report the variance of the empirical results.
- It would be better to provide the code and additional visualizations of the proposed methods.  For example, what did the adversarial samples look like? The importance of data-set memory. Since this paper has no source code, appendix, I would strongly recommend a revision by providing details of the proposed framework.


Side note: In general, I appreciate authors’ efforts by making a clear presentation on complex modules. However, this reviewer believes there are better ways to clearly present the method/paper rather than simply say we follow paper [xxx]. I would provide several examples.

- Related work (Sec 2.1-2.3) This paper simply presents different aspects such as domain adaptation, generalization, and semi-supervised. However, this reviewer feels quite redundancy in reading this, because this does not provide any insightful discussions about the ***relation** with your current paper.  It is just a simple combination of different recent papers….
- The comparison in Fig 1 and Tab 1 seem a bit redundant. I do think it is not hard to grasp the novelty in your setting. It would be much better to highlight why your setting is difficult and current approaches **does not work** well. E.g, several empirical results.
- The third concern is related to motivation. I know the proposed setting is generally uncommon. However, I could not feel a strong motivation of why it is important. Based on the experiments on the standard dataset, I feel it would be a bit unclear for me to understand the importance of your problem.
- Better explanations through better paper structures. I would strongly suggest a better illustration on the paper, for example why it is important to use EMA in your scenario? It is better to justify rather than saying based on paper [XXX].

---

> ### Author Response · Authors · 2023-09-06
> **Response to Reviewer  Ycg7**
>
> We thank the reviewer for providing constructive comments. In the following we provide detailed responses to these questions.
>
> **Q1: Why do we consider Wasserstein distance based adversarial samples?**
>
> **Response:** We appreciate the reviewer's inquiry regarding the threshold in Equation (9), and we would like to provide further clarifications. In our experiments, we set the threshold for Equation (9) in the same way as used in FixMatch. The threshold value plays a crucial role in our experiments. Its impact is two-folds: a) Lower threshold values tend to include more unlabeled samples, which can introduce noise information into the training process. This noise can potentially deteriorate the performance by introducing erroneous or misleading information from noisily pseudo-labeled samples. Conversely, higher threshold value limits the inclusion of unlabeled samples, reducing the volume of data for training, which can further lead to the model learning bias.
> **Q2: The threshold in Equation (9).**
>
> **Response:** We appreciate the reviewer's inquiry regarding the threshold in Equation (9), and we would like to provide further clarifications. In our experiments, we set the threshold for Equation (9) to follow the same threshold value as used in FixMatch, a well-established method in the field. The threshold value plays a crucial role in our experiments. Its impact is twofold: a) Lower threshold values tend to include more unlabeled samples, which can introduce noise information into the training process. This noise can potentially deteriorate the performance of the learned model by introducing erroneous or misleading information from noisy unlabeled samples. Conversely, higher threshold values limit the inclusion of unlabeled samples, reducing the availability of unlabeled data for training. This reduction can lead to a bias to the learned model due to the limited available unlabeled data jointly the training process.
>
> **Q3: The loss function in Equation (10).**
>
> **Response:** We appreciate the reviewer's question regarding Equation (10). To clarify, in Equation (10), we use the cross-entropy loss.
>
>  **Q4: Why using the data memory in Equation (2)?**
>
> **Response:** We appreciate the reviewer's question regarding the use of data memory in Equation (2). The rationale behind this approach is rooted in the adoption of a representative self-supervised learning method, specifically MoCo[1]. It utilizes a memory bank to store representations from recent batches of data. This memory bank serves as a source of negative samples in contrastive learning. The core idea is to maintain a queue of data representations, allowing for a dynamic selection of negative samples during the contrastive loss calculation. By incorporating this memory bank, MoCo enhances the quality and diversity of negative samples used in the loss function. This, in turn, promotes more effective and robust feature learning. The dynamic nature of the memory bank ensures that the model leverages a broader range of data representations, contributing to improved performance in self-supervised learning tasks.
>
> [1] Kaiming, He et al. (2020). “Momentum contrast for unsupervised visual representation learning.” In CVPR.
>
>  **Q5: Adding the variance of the empirical results.**
>
> **Response:** Thank you for the suggestion. We have included the discussion of variance in the revised submission. For more details, please refer to Appendix A.1.
>
>  **Q6: Providing the details of the proposed method and visualizing the generated adverasrial samples.**
>
> **Response:** Thank you for your valuable suggestions. We have made revisions to enhance the clarity of our proposed method. Additionally, we have included the core code of our method for your reference, which can be accessed through the following [anonymous link](https://drive.google.com/file/d/1lvUQjVhCeVR_EudulvVAnTU3qHgd2uLO/view?usp=sharing). Furthermore, we have provided visualizations of the generated data used in Equation (10). For more comprehensive information, please consult Appendix A.2.

---

> > ### Comment · Reviewer_Ycg7 · 2023-09-06
> >
> > Dear authors,
> >
> > Thank you for the responses. My technical concerns have been addressed.
> >
> > After checking the revised paper, I could found a better paper structure and polishing. While I still feel this paper can be improved for a better motivating story (but this is a minor point).

---

### Review · Reviewer_5cbU · 2023-08-20

**Summary Of Contributions:**

This paper studies a novel problem: semi-supervised (SS) single-source domain generalization (SDG). The key challenge vs. existing SDG works is the scarce labeled source data, making it hard to apply label-conditioned data augmentation to simulate diverse source data for domain generalization. The authors thus propose an algorithm combining *semi-supervised data augmentation* and *semi-supervised learning* to train the model. Experimental results on multiple multi-domain datasets show better accuracy vs. baseline approaches.

**Audience:**

Yes

**Broader Impact Concerns:**

No concerns.

**Claims And Evidence:**

Yes

**Requested Changes:**

**Major**

1. Please see the above weaknesses. I expect the authors to address them in the rebuttal or revised version.

2. Some technical details could be more precise or better motivated.

- Why is ss or ws needed in Eq (10)? Why do the authors only apply it to generated data rather than original data?

- The term "augmented" or "augmentation" is a bit overly used. For example, at the beginning of section 3.4. $D_g$ means *augmented* data, and the authors then mentioned strong *augmentation* and weak *augmentation*. I suggest that the authors differentiate these terms in the paper clearly.

3. In Figure 1, the test data of DG and SS-GD should contain multiple test domains like SDG and SS-SDG. In essence, DG trains a model that is expected to be generalizable to various test domains.

4. Please clearly describe how existing SDG methods are applied in the experiments. I suppose the authors only use the scarce labeled data to perform data augmentation. Nevertheless, a detailed experimental setup is recommended.

5. Not all contrastive learning approaches need a memory bank. Thus, the authors should modify section 3.2 to indicate which particular contrastive learning method was used clearly.

**Minor**

1. The paper has several typos and grammatical errors. For example:

- feature extractor pre-train --> feature extractor pre-training

- indicates the i-th unseen target domain data --> indicates data from the i-th unseen target domain.

- Here weak augmentation indicates augmentation --> please modify this sentence.

- Firstly, MPFM generates pesudo-labels utilizing the classifier F’s output --> I suppose this should be "C" not "F".

- Eq (6) --> I suppose it should be $G_g$ not $D_s$ in the CE loss.

2. The captions of the figures and tables should be self-contended. Thus, please expand the captions to mention the notations used in the figures and tables clearly (specifically, Figure 1 and Table 2). For instance, ERM (sup.) should be clarified.

**Strengths And Weaknesses:**

**Strength**

1. The paper studies a novel problem: semi-supervised (SS) single-source domain generalization (SDG).

2. The problem is well-motivated (good introduction section). The paper provides excellent figures and tables in the introduction section to compare related paradigms.

3. The paper clearly identifies the deficiency of existing solutions and proposes an algorithm that closely addresses the deficiency. (That is, the proposed solution is well-motivated and solid.)

4. The experiments contain multiple datasets and ablation studies.

**Weakness**

1. I hope the authors provide more discussions on the SS-SDG problem and outline potential solutions instead of sticking to only the data augmentation-based approaches. Concretely, as one of the first papers in SS-SDG, I encourage the authors to discuss the challenges and opportunities of SS-SDG broadly, which could strengthen the paper's impact. For instance, sect. 2.3 discusses multiple ways to SDG. Can other methods beyond data augmentation be applied or extended to SS-SDG? If not, what is the critical challenge, and how may we resolve them?

2. Returning to the proposed approach, while I buy the motivation of the proposed problem and the deficiency of existing SDG solutions, the proposed solution looks not exciting. Specifically, the proposed solution seems like a straightforward combination of self-supervised data augmentation and existing SSL algorithms (with a small extension to incorporate multiple pairs). This further leads to several doubts.

- First, while existing SDG methods use label-conditioned data augmentation, I suppose in the data augmentation community, there have been "learnable data augmentation methods" without the need to provide image labels as input. The current manuscript lacks a discussion on existing data augmentation methods (not necessarily proposed for SDG). I think the authors should provide a detailed survey and see if some existing methods are worth considering as baselines.

- Second, given that the main difference between SDG and SS-SDG is the scarce labeled data, I think a more direct solution is to first apply SSL methods in the source domain to recover the missing labels and then apply standard SDG approaches (maybe with some treatments to overcome noisy labels). Looking at Table 2 and Table 4, SSL methods alone, like FixMatch, have largely improved ERM. Applying SDG methods on top of the true and pseudo-labels should further improve the accuracy. This baseline approach should be compared.

3. I have doubts about whether a simple adversarial data augmentation can discover the variations among domains. I understand this may not be the core problem the manuscript needs to address. However, I am not fully convinced that adversarial data augmentation can create faithful augmented data reflecting actual domain variations.

4. According to Algorithm 1, the authors seem to generate a fixed set of $D_g$ at the end of Stage 1 and use it for Stage 2. However, since $G$ is still updated in Stage 2, I wondered whether $D_g$ should be regenerated or dynamically updated.

5. The authors should provide a stronger upper bound in Tables 2-4 to showcase the challenge of this research problem. Concretely, the current upper bound "ERM (sup.)" is simply a supervised model learned with the full source labeled data and without any treatment for domain generalization. In my opinion, the authors should provide an upper bound using SDG methods with full source labeled data. This will show the need to perform domain generalization. Otherwise, the current upper bound seems to suggest that if one can recover the missing labels in the source data, there is no need to perform domain generalization.

6. The paper only considers a single and relatively shallow model backbone ResNet 18. Stronger backbones like ResNet 50 or ViT should be considered.

7. The idea of "multi-pair" is not clearly motivated in the introduction. I suggest adding some descriptions to the introduction. Also, in Eq (10), there seem to be other ways to construct the pairs, for example, using s or w on $D_g$ rather than ss or ws. Also, the authors may consider pairing weakly augmented and strongly augmented $D_g$. It may be worth comparing them in Table 5.

---

> ### Author Response · Authors · 2023-09-06
> **Response to Reviewer 5cbU (Part 1)**
>
> We thank the reviewer for providing constructive comments. In the following we provide detailed responses to these questions.
>
> **Q1: Challenges, opportunitiess, and potential solutions of SS-SDG.**
>
> **Response:**We appreciate the reviewer's valuable suggestion to broaden our discussion on semi-supervised single domain generalization (SS-SDG). As one of the pioneering papers in this emerging field, we believe it's essential to provide a comprehensive perspective on the challenges and opportunities of SS-SDG. In addition to our primary focus on single domain generalization methods, e.g., most of them are data augmentation-based techniques. We acknowledge the existence of alternative strategies such as domain adaptation, domain generalization, and semi-supervised learning. These methods have been extensively studied in related domains and, indeed, have their merits. However, it's crucial to differentiate between these approaches and SS-SDG and discuss the limitations that make them less suitable for our specific problem.
>
> 1. Domain Adaptation: Domain adaptation aims to align the source and target domains to mitigate domain shift. While domain adaptation techniques have shown success in reducing domain discrepancy, they typically require access to labeled data in both source and target domains for effective training. In contrast, SS-SDG often involves scenarios where target domains are unavailable and source domain contains limited labeled data and plenty of unlabeled data, making traditional domain adaptation method less applicable.
>
> 2. Domain Generalization: Domain generalization, on the other hand, seeks to learn a model that can generalize across unseen domains during testing. While this aligns with the overarching goal of SS-SDG, domain generalization methods often assume access to multiple source domains during training to generalize effectively. SS-SDG, by definition, focuses on a single source domain with a limited labeled dataset, which distinguishes it from the domain generalization setting.
>
> 3. Semi-Supervised Learning: Semi-supervised learning is a powerful technique that leverages both labeled and unlabeled data to enhance model performance. However, in the context of SS-SDG, the primary challenge is often both the presence of a single source domain with limited labeled data and the domain gap between source and unseen target domains, making traditional semi-supervised learning methods less suitable without considering the domain divergency problem.
>
> 4. Drawbacks of Alternative Methods in SS-SDG: It's important to emphasize that the applicability of domain adaptation and domain generalization techniques may be limited in SS-SDG scenarios due to their assumptions of broader access to labeled data or multiple source domains. These assumptions may not hold in cases where we are dealing with a single source domain and a scarcity of labeled data in the source domain, which is a hallmark of SS-SDG problems.
>
> In conclusion, we acknowledge the broader landscape of potential methods such as domain adaptation, domain generalization, and semi-supervised learning, but emphasize that the unique characteristics of SS-SDG, particularly the challenge of dealing with a single source domain which contains limited labeled data and plenty of unlabeled data, necessitate tailored approaches. Our focus on data augmentation-based techniques in this paper reflects our effort to address these specific challenges and pioneer solutions in the emerging field of SS-SDG. Furthermore, our experimental results substantiate the validity of this approach.

---

> ### Author Response · Authors · 2023-09-06
> **Response to Reviewer 5cbU (Part 2)**
>
> **Q2: Discuss existing data augmentation methods.**
>
> **Response:**Data augmentation has become a widespread practice for enhancing the training of machine learning models, helping to mitigate overfitting and improve generalization. The fundamental concept behind data augmentation involves extending the original (data, label) pairs with new pairs, denoted as ($\mathbb{T}$(data), label), where $\mathbb{T}(\cdot)$ represents a transformation. In our specific problem, $\mathbb{T}(\cdot)$ serves as a means to simulate domain shifts. Existing data augmentation methods can be broadly categorized into three groups:
>
> 1. Hand-Crafted Transformations: This group encompasses the stacking of traditional image transformations within $\mathbb{T}(\cdot)$, such as random flips, rotations, and color augmentations. In addition to these conventional image transformations, there have been explorations into techniques like Mixup [1], which involves mixing instances at the image level. However, it's important to note that these transformations may have limitations in certain applications, potentially altering the associated labels.
>
> 2. Adversarial Learning Based Data Augmentation: Inspired by adversarial attacks [2], this approach involves perturbing images using sign-flipped gradients back-propagated from a classifier. This method is commonly used in single domain generalization to generate "challenging" images, making it more difficult for the classifier to make accurate judgments. However, a drawback is the increased computational cost associated with these techniques.
>
> 3. CNN Based Augmentation: In this category, Convolutional Neural Networks (CNNs) serve as the transformation function $\mathbb{T}(\cdot)$. The primary goal is to alter the styles of images, thereby enriching the diversity of the existing dataset. A notable limitation of these learnable image synthesis methods is that the newly generated images may not exhibit significant differences from images in other domains. This direction can be further divided into two subcategories:(1) Learnable Augmentation [3, 4, 5]: this direction focuses on learning an image translation model or employing pre-existing style transfer models like CycleGAN [6] or AdaIN [7] .(2) Non-Learnable Augmentation [8]: Here, T is randomly initialized with a Gaussian distribution at each iteration.
>
> It is worth mentioning that there exist learnable data augmentation methods [3, 4, 5] that do not require label information. However, these methods rely on style information from other domains, which is not available in the context of Semi-Supervised Single Domain Generalization (SS-SDG). Consequently, these learnable methods cannot be directly applied to our problem.
>
> [1] Hongyi, Zhangt et al. (2017). “Mixup: beyond empirical risk minimization.” In arXiv.
>
> [2] Han, Xu et al. (2020). “Adversarial attacks and defenses in images, graphs and text: A review.” In IJAC.
>
> [3] Nathan, Somavarapu et al. (2020). “Frustratingly simple domain generalization via image stylization.”  In arXiv.
>
> [4] Xiangyu, Yue et al. (2019). “Domain randomization and pyramid consistency: simulation-to-real generalization without accessing target domain data.” In ICCV.
>
> [5] Kaiyang, Zhou et al. (2020). “Learning to generate novel domains for domain generalization.” In ECCV.
>
> [6] Jun-Yan, Zhu et al. (2017). “Unpaired image-to-image translation using cycle-consistent adversarial networks.” In ICCV.
>
> [7] Xun, Huang et al. (2017). “Arbitrary style transfer in real-time with adaptive instance normalization.” In ICCV.
>
> [8] Zhenlin, Xu et al. (2020). “Robust and generalizable visual representation learning via random convolutions.” In arXiv.

---

> ### Author Response · Authors · 2023-09-06
> **Response to Reviewer 5cbU (Part 3)**
>
> **Q3: Suggests comparing a baseline approach of SDG methods by applying semi-supervised learning methods to assign pseudo labels for unlabeled source data.**
>
> **Response:**  We sincerely appreciate the reviewer's valuable suggestion regarding the introduction of new baselines. As per this strategy, denoted as ADA+PL and MEADA+PL, we utilize FixMatch to assign pseudo labels to the unlabeled source data. Following this, ADA and MEADA are applied to the complete dataset, encompassing both labeled and pseudo-labeled source data. For the performance of new baselines on PACS dataset are shown below. For more results on OfficeHome and DomainNet20, please refer to Table 2 and Table 3 in revised submission. In summary, our method consistently outperforms these additional baselines.
>
> |PACS-15 per class  |  A  | C |  P  |  R  |  Ave.  |
> |----|:--------:|:--------:|:-------:|:--------:|:------:|
>   | ERM (supervised) |70.9 |76.5 |43.3 |53.1 |60.7   |
>   | ADA (supervised) |72.43 |71.97 |44.63 |45.73 |58.70   |
>   | ERM |49.45 |50.84 |30.36 |25.25 |38.96   |
>   | ENT-MIN |54.15 |55.61 |38.08 |24.95 |43.20   |
>   | FixMatch |57.67 |69.13 |45.46 |42.17 |53.61   |
>   | ADA |47.87 |50.71 |30.31 |30.46 |40.08   |
>   | MEADA |48.79 |52.81 |34.23 |30.42 |41.56   |
>   | ADA+PL |59.31 | 65.69 | 43.23  | 39.31 | 51.89
>   | MEADA+PL |62.17 | 61.92 | 41.66 | 37.16 | 50.72   |
>   | Ours |**60.26** | **69.91** |**47.30** | **46.70** | **56.04**   |
>
>
>
>
>
>
> |PACS-25 per class|  A  | C |  P  |  R  |  Ave.  |
> |----|:--------:|:--------:|:-------:|:--------:|:------:|
>  |ERM (supervised) |70.9 |76.5 |43.3 |53.1 |60.7  |
>  |ADA (supervised) |72.43 |71.97 |44.63 |45.73 |58.70  |
>  |ERM |52.51 |54.05 |30.76 |23.61 |40.23  |
>  |ENT-MIN |56.10 |59.65 |36.99 |26.92 |44.91  |
>  |FixMatch |63.70 |68.27 |45.79 |42.86 |55.16  |
>  |ADA |51.71 |53.42 |30.33 |26.74 |40.55 |
>  |MEADA |51.97 |54.54 |32.27 |28.19 |41.74  |
>  |ADA+PL |59.63 |66.57 |44.43 |40.21 |52.71 |
>  |MEADA+PL |59.88 |65.92 |44.59 |41.44 |52.96 |
>  |Ours |**65.58**|**70.11**|**48.19** |**47.95**|**57.96**|
>
> **Q4:  The effectiveness of adversarial data augmentation in capturing domain variations.**
>
> **Response:**  We appreciate the reviewer's concern regarding the effectiveness of adversarial data augmentation in capturing domain variations. Adversarial data augmentation has been employed to enrich domain variations. Specially, it has indeed played an important role in single domain generalization. Several papers have demonstrated how adversarial data augmentation can enhance the generalization and robustness of models to domain shift. However, existing adversarial data augmentation techniques may not fully cover the entire unseen target domain variance. Without a specific direction or guidance towards the unseen target domains, it becomes challenging to create faithful augmented data that reflect the actual domain variations present in the unseen target domains. Even though adversarial data augmentation may not provide a comprehensive representation of the target domain's variations, it still plays a vital role in enhancing the robustness and generalization of deep learning methods. Both robustness and generalization can translate into improved performance. In conclusion, while we acknowledge the concerns about the limitations of adversarial data augmentation in fully characterizing target domain variations, we want to emphasize its role in enhancing the robustness and generalization of our approach. Additionally, we have observed significant performance improvements.
>
> **Q5:  Whether $D_g$ should be regenerated or dynamically updated in Stage 2?**
>
> **Response:** We appreciate the reviewer's keen observation regarding Algorithm 1. We would like to provide clarification that the set $D_g$ in Stage 1 is dynamically updated. This dynamic update occurs because Stage 1 involves self-supervised learning as part of our proposed data augmentation approach. Consequently, both the G (generator) and $D_g$ (generated data) are continually updated during Stage 1. However, we do not update $D_g$ at Stage 2 for two key reasons: firstly, our data augmentation method is integrated with self-supervised learning, which is not involved in Stage 2. This differentiation in the learning processes between stages leads to the decision not to update $D_g$ in Stage 2. Secondly, drawing from the experimental insights gained from existing adversarial data augmentation-based single domain generalization approaches, data augmentation is typically performed during the early training epochs. Consequently, updating $D_g$ during Stage 2 may not align with the established practices in this domain.

---

> ### Author Response · Authors · 2023-09-06
> **Response to Reviewer 5cbU (Part 4)**
>
> **Q6: Providing a stronger upper bound using SDG methods with full source labeled data**
>
> **Response:** We sincerely appreciate the valuable suggestion from the reviewer. We have evaluated SDG methods, specifically ADA, using the complete source labeled data, denoted as ADA (supervised). Below, we present the averaged performance of this baseline on the PACS, OfficeHome, and DomainNet20 datasets. For more detailed results, please refer to Tables 1, 2, and 3 in the revised submission. In summary, our findings indicate that even when providing full labeled source domain data to representative SDG methods, they do not consistently gain a significant advantage over ERM when facing challenging datasets such as PACS, OfficeHome, and DomainNet20. This observation potentially underscores the inherent challenges in our SS-SDG problem.
>
> |Method |  PACS  |OfficeHome | DomainNet20|
> |----|:--------:|:--------:|:-------:|
>  |ERM (supervised) |60.70 |50.36 |58.93 |
>  |ADA (supervised) |58.70 |51.60 |59.54 |
>
> **Q7: Considering stronger backbone models such as ResNet 50 or ViT.**
>
> **Response:**  We highly appreciate the valuable suggestion provided by the reviewer. We have conducted evaluations for all methods using ResNet50 on the DomainNet20 dataset. During this evaluation, we designated the Clipart domain as the source domain, while the other domains were considered as unseen target domains. Our findings consistently demonstrate that our methods significantly outperform all baselines, with the exception of the two upper bounds, which leverage full labeled source domain data for model training.
>
>  |Method |ERM (supervised) |ADA (supervised) |ERM |ENT|FixMatch | ADA| MEADA| ADA+PL| MEADA + PL| Ours|
> |----|:-----:|:-----:|:----:|:----:|:----:|:----:|:----:|:----:|:----:|:----:|
>  |Clipart |68.14 |69.98 |55.75 |59.85 |63.62 | 56.97| 57.11 |55.84 | 57.31 | 65.82|
>
>
> **Q8: Providing clearer motivation for the "multi-pair" and exploring alternative ways to construct pairs in Equation (10).**
>
> **Response:** We greatly appreciate the valuable suggestions provided by the reviewer. We have taken the feedback into consideration and made the necessary revisions to improve the clarity and understanding of our approach. Please refer to Section 3.4 for the updated details.
>
> We have conducted experiments to compare different pair construction methods, specifically using $s$ or $w$ on $D_g$ versus $ss$ or $ws$. As shown the experimental results in the following table, it indicates that $ss$ or $ws$, the design used in our method, consistently outperforms $s$ or $w$. This suggests that $ss$ or $ws$ is a better choice for forming multi-pairs.
>
> |PACS |Using $D_g^{ss}$ and $D_g^{ws}$? |A |C |P |S |Avg|
> |----|:--------:|:--------:|:-------:|:-------:|:-------:|:-------:|
> |Ours |  |59.76 |68.45 |46.71 |45.98 |55.23|
>  |Ours |$\checkmark$ |60.26 | 69.91 | 47.30 | 46.70 | 56.04|
>
>
> The fundamental concept behind FixMatch is to create both weakly and strongly transformed samples. In this process, weakly transformed data plays a pivotal role in generating pseudo-labels. If these pseudo-labels are considered reliable, they are then used as label information for the strongly transformed data, facilitating the model's learning process. It's worth emphasizing that both weakly and strongly transformed data stem from the same source, ensuring that they share identical semantic information. Furthermore, when compared to strongly transformed data, weakly transformed data offers a higher likelihood of generating accurate pseudo-labels due to its inherent characteristics. In contrast, pairing weakly augmented and strongly augmented data from $D_g$ = {$D_g^s$, $D_g^w$\} may lead to a deterioration in the quality of pseudo-labels when compared to using $D_s^w$ directly. Additionally, this approach of pairing weakly and strongly augmented data from$D_g$ = {$D_g^s$, $D_g^w$\}results in only one pair, which inherently possesses less diversity and consequently results in lower-quality pseudo-labels.

---

> ### Author Response · Authors · 2023-09-06
> **Response to Reviewer 5cbU (Part 5)**
>
> **Q9: Typos and grammatical errors.**
>
> **Response:** We appreciate the reviewer for pointing out the typos and grammatical errors. We have thoroughly revised these issues in the revised submission.
>
> **Q10: Clearly present the paper.**
>
> **Response:** We appreciate the reviewer for pointing out how to improve the quality of the paper. For example, make the figure 1 more clear  to describe the problem, giving details of the experiment, clearing present the contrastive learning used in paper. We have carefully follow these guidlines to revise the paper. More detials please refer to the  revised submission.

---

> ### Comment · Action_Editors · 2023-09-25
>
> Can you post your "official recommendation" when you are available for this?

---

### Review · Reviewer_gbFc · 2023-08-25

**Summary Of Contributions:**

The paper studied a new setting, semi-supervised single domain generalization (SS-SDG), where the model is trained from one partially labeled source domain and generalizes to multiple unknown target domains. An effective framework was proposed for this new setting that consists of three key components: 1) pre-train the model based on contrastive learning; 2) diversify the source domain data by adversarial data augmentation; 3) apply the multi-pair FixMatch regularization in training. The proposed framework outperforms baselines in experiments on three datasets: OfficeHome, PACS and DomainNet20.

**Audience:**

Yes

**Claims And Evidence:**

Yes

**Requested Changes:**

1. I think it is better to add more explanations on why the proposed SS-SDG setting is important to be studied (i.e., weakness 2).
2. I suggest to add more explanations on how to ensure fairness comparison with baselines (i.e., weakness 3).
3. Table 1 seems to be unnecessary given the Figure 1. It may be better to keep Figure 1 only.

**Strengths And Weaknesses:**

- Strengths
1. The paper is well written and presented. In particular, the relationship between the proposed setting and prior settings is clearly explained and presented, e.g., Figure 1.
2. The paper presented a new setting SS-SDG, where the single source domain contains unlabeled data and limited labeled data, which is expected to generalize to multiple unknown target domains. The setting seems to be more difficult than previous SDG setting.
3. The proposed method achieved good experimental performance when compared with the baselines. The authors also included an ablation study to examine the effectiveness of the components within the proposed method, which is quite important for checking the proposed method in-depth.

- Weaknesses
1. The proposed method is quite artificial and heuristics, more like a combination of several existing techniques, i.e., contrastive learning, adversarial data augmentation and FixMatch regularization, without any theoretical motivation/guarantee.
2. After reading the paper, I feel not so convinced by why the proposed SS-SDG setting is important. It would be more clear if additional explanations are given on this point, e.g., motivations of the setting, some real-world examples.
3. I have a concern of how to ensure the fairness in experiments when comparing the proposed method with baselines. It seems to me that the proposed method contains many key techniques, e.g., adversarial data augmentation, contrastive learning. I'm wondering if the baselines also had such techniques in the empirical comparison, e.g., whether the baselines were conducted on top of the same degree of data augmentation as that in the proposed method.

---

> ### Author Response · Authors · 2023-09-06
> **Response to Reviewer gbFc (Part 1)**
>
> We thank the reviewer for providing constructive comments. In the following we provide detailed responses to these questions.
>
> **Q1: Contributions of our work and the distinctions between our methods and existing techniques.**
>
> **Response:** We sincerely appreciate the valuable feedback provided by the reviewer. Our work's primary contribution does not hinge on the entire framework; instead, we focus on designing novel components, to address the challenging problem of semi-supervised single domain generalization (SS-SDG). Unlike traditional data augmentation techniques such as those used by FixMatch (e.g., random cropping and flipping), our approach employs a "loss-driven auto-augmentation" strategy. This approach aims to introduce new samples to enhance the robustness and generalization ability of the learned model. It's important to note that FixMatch is constrained by a limited number of augmentation types and parameter combinations. Additionally, the augmented data generated by FixMatch is not adversarial in nature. In the second stage of our method, we utilize a technique called CADA on samples weakly and strongly generated by adversarial learning. This process results in a broader data diversity, denoted as $D_{g}^{ws}$ and $D_{g}^{ss}$, as opposed to the original $D_{g}^{s}$. Our approach demonstrates a clear empirical improvement, as evidenced by a 2.4% accuracy gain in Table 4 when comparing "FM" to "Ours." Another important distinction lies in the data generation methods used. Traditional adversarial data generation techniques such as ADA and MEADA rely on label information. In contrast, our CADA method is label-agnostic, which allows for the effective scaling up of training data using a large amount of unlabeled data. This unique feature contributes to the strength and versatility of our approach.
>
> **Q2: Motivation and Importance of SS-SDG Setting.**
>
> **Response:** We acknowledge the reviewer's request for a practical example of SS-SDG and would like to provide an illustrative case in the context of face recognition. In this scenario, the offline model is equipped with labeled data, typically acquired in controlled settings or through prior registration procedures. However, as the system encounters new users, especially in open and dynamic environments like surveillance settings, there may not be a prior registration step, leading to a significant portion of the data being devoid of labels. In this context, the term "source" data remains consistent and unchanging, representing the labeled dataset gathered during the offline phase. On the other hand, the "target" data exhibit variability across diverse scenarios, such as indoor and outdoor environments. This introduces a notable challenge, as the offline data are enriched with labels, while the data collected in real-time, which forms the "target" data, are predominantly unlabeled. To summarize, this practical application of SS-SDG in face recognition confronts the task of effectively handling both labeled offline data and the predominantly unlabeled real-time data, with the goal of learning a generalizable and robust model for face recognition.

---

> > ### Author Response · Authors · 2023-09-06
> > **Response to Reviewer gbFc (Part 2)**
> >
> > **Q3: How to ensure the fairness in experiments?**
> >
> > **Response:** We genuinely value the constructive feedback offered by the reviewer. To ensure the fairness of our experimental comparisons between the proposed method and baselines, it's essential to consider the unique characteristics of our problem, i.e., Semi-Supervised Single Domain Generalization (SS-SDG). Here's how we address the concern:
> >
> > 1. Data Augmentation: Data augmentation is indeed a widely used technique in the context of single domain generalization (SDG). However, our SS-SDG problem presents distinct challenges that make standard data augmentation less suitable. In our experiments, we compare the proposed method against strong baseline methods, specifically ADA (Adversarial Data Augmentation) and MEADA (Modified Entropy-based Adversarial Data Augmentation), which are representative data augmentation techniques for SDG. This allows us to evaluate the effectiveness of our proposed method in the context of SS-SDG while providing a meaningful point of comparison.
> >
> > 2. Semi-Supervised Learning: Semi-supervised learning is another strategy that can be applied to address SS-SDG problems. However, it often falls short in addressing the domain gap issue, which is a crucial aspect of our SS-SDG problem. To evaluate our proposed approach comprehensively, we selected FixMatch, a widely recognized and representative semi-supervised learning method, as a strong baseline. This choice allows us to assess how our method compares against a well-established approach in the semi-supervised learning domain.
> > 3. Addressing Fairness: To further address the concern of fairness in our experiments, we introduced additional baselines, namely ADA-FixMatch and MEADA-FixMatch. These baselines combine the data augmentation techniques (ADA and MEADA) with the semi-supervised learning method (FixMatch), aligning them more closely with the key techniques used in our proposed method. This addition allows for a fairer comparison that directly addresses the question of whether the proposed adversarial data augmentation and contrastive learning techniques provide significant improvements over standard data augmentation and semi-supervised learning approaches.
> >
> > For the additional baselines, i.e., ADA+FixMatch and MEADA+FixMatch, performances on PACS dataset are shown below. For more results on OfficeHome and DomainNet20, please refer to the Table 2 and Table 3 in revised submission. In summary, our method consistently outperforms these additional baselines.
> >
> > |PACS-15 per class |  A  | C |  P  |  R  |  Ave.  |
> > |----|:--------:|:--------:|:-------:|:--------:|:------:|
> >   | ERM (supervised) |70.9 |76.5 |43.3 |53.1 |60.7   |
> >   | ADA (supervised) |72.43 |71.97 |44.63 |45.73 |58.70   |
> >   | ERM |49.45 |50.84 |30.36 |25.25 |38.96   |
> >   | ENT-MIN |54.15 |55.61 |38.08 |24.95 |43.20   |
> >   | FixMatch |57.67 |69.13 |45.46 |42.17 |53.61   |
> >   | ADA |47.87 |50.71 |30.31 |30.46 |40.08   |
> >   | MEADA |48.79 |52.81 |34.23 |30.42 |41.56   |
> >   | ADA+FixMatch |60.02 |69.44 |45.36 |39.24 |53.51   |
> >   | MEADA+FixMatch |59.99 |69.65 |46.41 |40.55 |54.15   |
> >   | Ours | **60.26** | **69.91** | **47.30** | **46.70** | **56.04**   |
> >
> > |PACS-25 per class|  A  | C |  P  |  R  |  Ave.  |
> > |----|:--------:|:--------:|:-------:|:--------:|:------:|
> >  |ERM (supervised) |70.9 |76.5 |43.3 |53.1 |60.7  |
> >  |ADA (supervised) |72.43 |71.97 |44.63 |45.73 |58.70  |
> >  |ERM |52.51 |54.05 |30.76 |23.61 |40.23  |
> >  |ENT-MIN |56.10 |59.65 |36.99 |26.92 |44.91  |
> >  |FixMatch |63.70 |68.27 |45.79 |42.86 |55.16  |
> >  |ADA |51.71 |53.42 |30.33 |26.74 |40.55 |
> >  |MEADA |51.97 |54.54 |32.27 |28.19 |41.74  |
> >  |ADA+FixMatch |59.73 |68.97 |44.31 |40.58 |53.40 |
> >  |MEADA+FixMatch |61.12 |70.10 |47.41 |40.81 |54.86 |
> >  |Ours  |**65.58**|**70.11**|**48.19** |**47.95**|**57.96**|
> >
> > **Q4: About the Table 1 and Figure 1**
> >
> > **Response:** We appreciate the valuable suggestion from the reviewer. In the revised submission, we have retained only Figure 1.

---

> > > ### Comment · Reviewer_gbFc · 2023-09-17
> > > **Post-rebuttal comment**
> > >
> > > After checking the responses and the updated paper, my concerns were appropriately addressed. Thank the authors for the rebuttal!

---

> > ### Comment · Action_Editors · 2023-09-29
> >
> > > In this context, the term "source" data remains consistent and unchanging, representing the labeled dataset gathered during the offline phase.
> >
> > Is this consistent with what was plotted in Fig 1 about SS-SDG where the source domain looks partially labeled?

---

> > > ### Comment · Action_Editors · 2023-09-29
> > >
> > > I found the motivating example is not exactly the same in the rebuttal and in the revised submission (as I didn't find the mentioned sentence in the submission). I guess the submission is correct and the rebuttal is a bit misleading. Anyway, please double check and guarantee the description in the camera-ready version is correct. Although I recommended "accept as is", you may revise your submission as you see fit. Just I don't require you go for a minor revision if you think it's unnecessary.

---

> > > > ### Author Response · Authors · 2023-09-30
> > > >
> > > > Dear Action Editor,
> > > >
> > > > Thank you very much for your comments.
> > > >
> > > > Yes. The motivating example in our revised submission is correct, while the one in the rebuttal is misleading. The source domain in the face recognition example should be partially labeled, as illustrated in Figure 1.
> > > >
> > > > We will carefully check the final version of our paper and make sure the description is correct.
> > > >
> > > > Thanks again for your suggestions!
> > > >
> > > > Best,
> > > >
> > > > Authors

---

### Decision · Action_Editors · 2023-09-29

**Recommendation:** Accept as is

**Comment:**

The submission studied a new problem setting called semi-supervised single domain generalization which is conceptually interesting and practically important (by the way, Figure 1 is especially nice for understanding the new setting). A learning method was then proposed to solve the above problem as a combination of three previously existing techniques (contrastive learning, adversarial data augmentation, and FixMatch regularization). The author rebuttals successfully addressed most of the concerns from the reviewers. Three reviewers (including an emergency reviewer) recommended acceptance while one reviewer was not responsive and did not submit his/her official recommendation. Thus, I think we should accept the submission for publication at TMLR.

**Audience:**

Yes

**Claims And Evidence:**

Yes